# PARALLEL-R1: TOWARDS PARALLEL THINKING VIA REINFORCEMENT LEARNING

**Tong Zheng**[1,2], **Hongming Zhang**[1], **Wenhao Yu**[1], **Xiaoyang Wang**[1], **He Xing**[1], **Ruipeng Dai**[3],
**Rui Liu**[2], **Huiwen Bao**[4], **Chengsong Huang**[5], **Heng Huang**[2], and **Dong Yu**[1]

[1]Tencent AI Lab Seattle
[2]University of Maryland, College Park
[3]University of North Carolina at Chapel Hill
[4]City University of Hong Kong
[5]Washington University in St. Louis

## ABSTRACT

Parallel thinking has emerged as a novel approach for enhancing the reasoning capabilities of large language models (LLMs) by exploring multiple reasoning paths concurrently. However, activating such capabilities through training remains challenging. Existing methods mainly rely on supervised fine-tuning (SFT) over synthetic data, which encourages teacher-forced learning rather than exploration and generalization. To address this issue, we propose **Parallel-R1**, the first reinforcement learning (RL) framework that instills parallel thinking for complex real-world reasoning tasks, e.g., mathematical reasoning task, beyond synthetic tasks. Our framework employs a progressive curriculum that addresses the cold-start problem in training parallel thinking with RL. We first use SFT on prompt-generated trajectories from easier tasks to instill the parallel thinking behavior, then transition to RL to explore and generalize this skill on harder problems. Experiments on various math benchmarks, including MATH, AMC23, and AIME, show that Parallel-R1 successfully elicits parallel thinking, leading to 8.4% accuracy improvements over the sequential thinking model trained directly on difficult tasks with RL. Further analysis reveals a distinct shift in the model's thinking patterns: in the early stage, it utilizes parallel thinking as an exploration strategy, while in the later stage, it employs this ability for multi-perspective verification. Most significantly, we validate parallel thinking as a **mid-training exploration scaffold**, where this intermediate phase unlocks a higher performance ceiling after RL, yielding a **42.9%** improvement over the sequential RL baseline.

## 1 INTRODUCTION

Google's Gemini (Comanici et al., 2025) recently credited its success at the International Mathematical Olympiad (IMO) in part to a novel capability: parallel thinking (Luong & Lockhart, 2025). This approach, as exemplified in Figure 1 (top), parallelizes multiple independent reasoning branches before synthesizing them into a single conclusion. Although the underlying mechanism remains unknown, this success highlights the value of parallel thinking for future LLM development. In cognitive science, it is well established that humans depend on such parallel exploration, considering multiple possibilities before converging on coherent conclusions. This fosters divergent thought and helps reduce the risk of premature closure to suboptimal solutions (Johnson-Laird, 1994; Sowden et al., 2019). Moreover, parallel thinking provides such benefits by leveraging the concurrent execution capability of modern GPUs to explore multiple reasoning paths in parallel (Yang et al., 2025b). Inspired by these, this work investigates how to effectively incorporate parallel thinking into LLMs.

Despite its promise, activating parallel thinking remains challenging, as the sequential nature of current LLMs inherently constrains this ability. While test-time strategies (Yao et al., 2023; Wang et al., 2022; Brown et al., 2024; Zhang et al., 2024; Hsu et al., 2025; Rodionov et al., 2025; Fu et al., 2025; Xiong et al., 2025a) can elicit such behavior, they only work during inference without internalizing

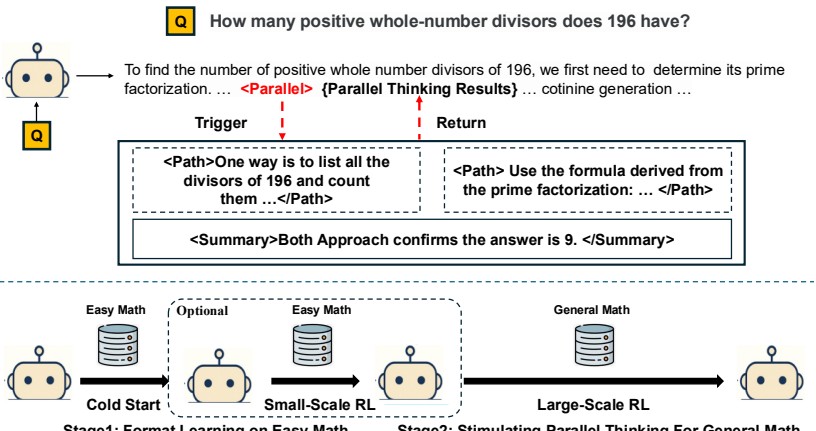

Figure 1: **An overview of Parallel-R1**. (Top) During inference, the model generates in a standard autoregressive fashion until emitting a special `<Parallel>` tag. At that point, it spawns multiple threads to explore different reasoning branches or perspectives, then summarizes their outputs into one conclusion. Next, these contents are merged back into the main context, and autoregressive generation continues. This cycle may repeat several times before the model arrives at the final answer. (Bottom) Parallel thinking is activated through a progressive multi-stage training approach. Intuitively, the approach first equips the model with parallel thinking ability on easy math problems and then progressively extends it to more general and difficult problems through reinforcement learning.

the capability. This has spurred growing interest in permanently instilling this capability through parallel-aware training. However, current training-based approaches fall short of this goal. Methods based on supervised fine-tuning (SFT) (Yang et al., 2025b; Macfarlane et al., 2025; Chen et al., 2025) (i) depend on costly pipelines to curate high-quality trajectories; (ii) rarely yield performance gains beyond faster inference; and (iii) achieve parallelism via superficial teacher-forcing rather than free exploration. Consequently, while models can replicate known patterns, their capability to generalize the underlying parallel thinking strategies is severely limited.

In contrast, reinforcement learning (RL) offers a more scalable approach to instilling parallel thinking in LLMs through exploration. However, since current autoregressive LLMs are not exposed to parallel thinking behaviors during pre-training or SFT, they are unable to produce such trajectories natively. As a result, a cold-start stage becomes essential for seeding this capability with supervised signals. This stage requires a large-scale, high-quality dataset to teach basic formats, which is scarce for complex real-world problems and difficult to synthesize. This explains why successful applications of RL for parallel thinking have solely been confined to narrow, synthetic domains, such as the CountDown task (Pan et al., 2025). Additionally, the optimal reward function for reinforcing parallel thinking with RL remains an open question. Relying solely on outcome-based rewards drives models toward shortcuts that bypass genuine parallel thinking, whereas enforcing structure-based rewards may lead to parallelism even when it is unnecessary. Furthermore, the strategic role and underlying mechanisms of parallel thinking remain insufficiently understood—for instance, how reasoning behaviors evolve and benefit training—thereby limiting our ability to fully harness their potential.

To address these challenges, we present **Parallel-R1**, the first RL framework for instilling parallel thinking in LLMs on general mathematical reasoning tasks. First, to address the model's lack of format knowledge for parallel thinking, we start our training process with an SFT stage on easy math tasks. A key observation is that simple prompting already yields high-quality trajectories on such problems (see Table 1), enabling the construction of the *Parallel-GSM8K* dataset. Building on this finding, we design a progressive curriculum: SFT on easy problems to learn basic formats, followed by RL on harder tasks to generalize the capability. Second, to tackle the critical challenge of reward design, we propose and investigate multiple reward schemes and find an effective alternating reward strategy, which switches between an outcome-based (accuracy) reward and a reward that encourages parallel thinking behaviors within fixed windows achieves a superior balance between performance and parallel thinking behaviors. Finally, to shed light on the black-box role of parallel thinking, we analyze the model's learned parallel thinking behavior and uncover a strategic evolution: parallel thinking shifts from computational exploration in early stages to multi-perspective verification in

later stages. This further motivates our concept of parallel thinking as a mid-training scaffold, which empirically boosts performance, reaching 25.6% on AIME25. We explore these contributions across both autoregressive and multiverse variants to provide robust insights into architectural design.

In summary, our core contributions can be concluded as follows:

- We propose Parallel-R1, the **first RL framework** to learn parallel thinking on general mathematical reasoning tasks, enabled by our **progressive training curriculum** and **dedicated reward design**.
- We delve deep into the learning dynamics, revealing that the target of parallel thinking evolves from **exploration to verification**. Additionally, we identify the concept of parallel thinking as a **mid-training exploration scaffold**, leading to a 42.9% gain over the sequential baseline after RL.
- We provide comprehensive experiments to show the consistent gains of our approach across various benchmarks. Our ablations further offer practical insights into reward and architectural design.

## 2 Related Work

### 2.1 Parallel Thinking

Parallel thinking has recently emerged as an active area of research. A common brute-force strategy is to spawn multiple independent trajectories at the very beginning and join their outcomes only at the end (Brown et al., 2024; Wang et al., 2022), or to exchange thoughts at fixed intervals (Rodionov et al., 2025; Hsu et al., 2025). Obviously, such schemes lack adaptivity as the points of branching and aggregating are dictated by a pre-defined schedule, not conditioned on the intermediate progress of the thinking process itself. To achieve finer-grained control, methods such as Monte Carlo Tree Search (Zhang et al., 2024) and Tree of Thoughts (Yao et al., 2023) offer more nuanced parallelism; however, they are still guided by hand-crafted heuristics based on external verifiers. Recent work (Pan et al., 2025; Yang et al., 2025b) strives for adaptive parallelism through RL or SFT. However, these studies either (i) focus mainly on efficiency—losslessly converting a single long CoT into an adaptive parallel form via SFT, which limits the discovery of new reasoning patterns, or (ii) verify RL only on synthetic tasks like Countdown. In this work, we argue that learning parallel thinking via RL is a more generic and promising direction: it uncovers novel, highly adaptive reasoning behaviors, leading to improved performance beyond the "lossless transformation" paradigm of Yang et al. (2025b). To this end, we propose Parallel-R1, the first RL framework that enables adaptive parallel thinking ability for general mathematical reasoning tasks.

### 2.2 Improving Reasoning via RLVR

Reinforcement Learning with Verifiable Rewards (RLVR) optimizes language models via RL using outcome-based, automatically checkable rewards, eliminating the need for trained reward models and the corresponding reward hacking. Recent advances have demonstrated the effectiveness of RLVR in diverse domains (Guo et al., 2025; Wang et al., 2025a; Huang et al., 2025b; Wang et al., 2025b; Liu et al., 2025; 2026). In parallel, a growing body of work aims to make RLVR more efficient and stable, proposing new training paradigms such as self-play (Zhao et al., 2025; Huang et al., 2025a) and test-time RL (Zuo et al., 2025; Zhou et al., 2025), as well as more robust RL algorithms (Yu et al., 2025; Yue et al., 2025; Wang et al., 2025c). However, existing autoregressive LLMs do not inherently possess parallel thinking abilities; therefore, we cannot directly instill such parallel thinking ability into LLMs through those RLVR approaches. To this end, we present the first RL framework that extends RLVR to parallel thinking on general reasoning tasks (e.g., mathematical problem solving).

## 3 Learning Parallel Thinking via Reinforcement Learning

Previous approaches to training parallel thinking in real-world tasks have primarily relied on SFT (Yang et al., 2025b; Macfarlane et al., 2025; Chen et al., 2025), a paradigm that suffers from several key limitations. To overcome these issues, we propose **Parallel-R1**, the first RL framework that enables adaptive parallel thinking for complex reasoning tasks. Our key idea is to bootstrap with simple tasks to generate cold-start data, and then use RL to transfer parallel thinking to harder

problems, thus avoiding complex data generation for challenging cases. Within this framework, we investigate two complementary settings for learning parallel thinking: one without architectural modifications and another with explicit architectural changes. In the latter, we modify both attention masks and position ids to prevent cross-attention between independent paths following Multiverse.

## 3.1 Formulation of Parallel Thinking Behaviors

When solving complex problems, humans often encounter moments of confusion or uncertainty, which we define as "critical steps" within a reasoning chain (Wang et al., 2025c; Li et al., 2025). At these points, engaging in parallel thinking allows us to explore multiple candidate solution paths simultaneously, thereby facilitating convergence toward a more reliable and higher-quality conclusion. Motivated by this paradigm, we formalize parallel thinking in LLMs as a two-stage process:

1. **Exploration**: When the model identifies a critical step, it temporarily suspends the main path and launches a multi-branch search, generating $N$ independent reasoning branches simultaneously.
2. **Summary**: After exploration, the model aggregates the outcomes from all branches, extracts their unique key insights, and resolves conflicts to arrive at the most promising final conclusion. Subsequently, it automatically resumes the main reasoning path and inputs this summarized conclusion.

We allow the model to interleave parallel thinking by repeating these two stages whenever needed during the generation process. An illustration of one such unit is provided in Figure 1 (Top). To realize this behavior, we introduce three control tags: `<Parallel>...</Parallel>`, `<Path>...</Path>`, and `<Summary>...</Summary>`, corresponding to the exploration phase, the isolation of independent reasoning paths, and the summarization of all parallel thinking paths, respectively. Using these tags, we further define the workflow during inference as follows:

**Workflow at Inference Phase**  During inference, our trained model dynamically activates parallel thinking behaviors as follows. It begins with autoregressive generation along the main reasoning path. When a `<Parallel>` token is predicted, the model pauses the main reasoning thread and concurrently generates multiple reasoning threads within separate `<Path>...</Path>` blocks. Once all parallel threads are completed, their outputs are automatically consolidated into a concise `<Summary>...</Summary>` block, which integrates insights from diverse perspectives. The summarized context is then used to resume and complete the main reasoning thread. This adaptive and dynamic inference procedure effectively exploits parallelism to enhance reasoning performance.

## 3.2 The Lightweight Data Pipeline for Parallel Thinking Cold-Start

Due to the sequential nature of both LLMs and humans, collecting high-quality parallel thinking data is a significant challenge. Even though humans think in the parallel fashion, they will summarize and only say/write the summarization. Thus, such data is extremely rare in the natural distribution. Existing approaches (e.g., Yang et al. (2025b)) exploit the inherent parallelism of long CoTs. However, they rely on

Table 1: Comparison of the ratio of parallel thinking data generated by DeepSeek-R1-0528-Qwen-3-8B on the DAPO and GSM8K datasets under the same prompt templates and sampling settings.

| Data | # Samples | Parallel Thinking Format (%) |
|------|-----------|------------------------------|
| GSM8K | 7472 | **83.7** |
| DAPO | 17916 | 0.0 |

complex, multi-stage data pipelines to rewrite long CoT trajectories to parallel reasoning demonstrations, avoiding human annotations. While highly effective for imitation learning, such rewritten traces do not originate from native parallel reasoning and therefore pre-shape the model with teacher-induced reasoning behaviors. This makes them fundamentally unsuitable as cold-start supervision for our scientific goal: RL induce parallel-thinking behaviors through exploration.

To address this issue, our key observation is that **while a lightweight prompting approach struggles to generate high-quality parallel thinking data for complex problems from the DAPO dataset, it proves highly effective for simpler tasks from GSM8K**, as showcased in Table 1. Based on this finding, we propose a simple data pipeline that prompts LLMs with detailed instructions to construct a large-scale, high-quality corpus for easier problems. Specifically, we utilized DeepSeek-R1-0528-Qwen-3-8B as a data generator, seeded with the 7,473 samples of the GSM8K training set, to generate high-quality parallel thinking trajectories. Next, we extracted the non-thinking parts to serve as gold annotations, creating our cold-start dataset called *Parallel-GSM8K*.

As the structured model variant (described in Section 3.4) involves architectural modifications, strict adherence to the format is required for successful training. Therefore, to ensure the quality and alignment of this corpus, we perform an additional filtering step, a Parallel Thinking Format Check, which is implemented by Algorithm 1. Crucially, this dataset is used to instill the basic format of parallel thinking, serving as a foundation for RL to generalize this ability to more complex problems.

## 3.3 LEARNING PARALLEL THINKING VIA RL IN AUTOREGRESSIVE MODELS

We further explore strategies to enable parallel thinking without modifying the model architecture.

### 3.3.1 REINFORCEMENT LEARNING ALGORITHMS

We use Group Relative Policy Optimization (GRPO) (Shao et al., 2024) as our RL algorithm. The detailed background about GRPO is further provided in Appendix C. The rollout process follows a multi-turn interactive framework, where the LLM alternates between sequential generation, parallel exploration, and sequential summarization, with the detailed process being described in Section 3.1.

### 3.3.2 CURRICULUM TRAINING AND REWARD MODELING

During the training of Parallel-R1, a progressive curriculum is applied, involving three stages: (i) Cold-Start Stage (i.e., SFT on Easy Math); (ii) RL on Easy Math, and (iii) RL on General Math.

**Cold-Start Stage.** To initialize the RL actor, we first apply SFT on the curated Parallel-GSM8K dataset introduced in Section 3.2. This cold-start stage addresses the model's lack of format knowledge, equipping it with a basic capability to generate outputs in the correct parallel thinking format.

**RL on Easy Math.** After the cold-start stage with SFT, the model already possesses the basic ability to generate special tags for parallel thinking, but the behavior is not stable since these tags have never appeared during pre-training. To address this issue, we further perform small-scale RL to enhance the format learning, where we use the same question set following our cold-start stage and use GRPO (Shao et al., 2024) for our RL training. To ensure both parallel ratio and accuracy, the final reward format in this stage is defined as: $R_{final} = R_{\langle \text{Parallel} \rangle} \ \& \ R_{\text{acc}}$. Here, the Accuracy Reward ($R_{\text{acc}}$) evaluates the correctness of the final response, while the Parallel Reward ($R_{\langle \text{Parallel} \rangle}$) incentivizes the model to use parallel thinking in its generation. This reward structure is designed to be binary and strict: a positive reward of +1 is given only if the generated output contains at least one parallel thinking unit **AND** the final answer is correct. Otherwise, the model receives a penalty of -1.

**RL on General Math.** After training stages on easy math, the model should be stable enough to generate control tags and produce outputs in the correct parallel thinking format if needed, but it still struggles with more challenging mathematical tasks. To address this, we further apply RL to general math datasets, thereby generalizing the model's parallel thinking ability beyond simple cases.

Specifically, we use the accuracy reward ($R_{\text{acc}}$) as our sole reward in this stage. This is because the primary goal of this stage is to improve task performance. For the seed problems, we select the widely used DAPO dataset. Finally, models trained from this stage yield our *Parallel-Seen* variant.

## 3.4 ELICITING PARALLEL THINKING VIA RL IN MULTIVERSE MODELS

In the previous section, we explored an RL framework that trains models to use parallel thinking without modifying their autoregressive architecture. However, this approach, which we refer to as Parallel-Seen, does not explicitly isolate reasoning paths during training. Therefore, both forward computation and backward gradients interfere across paths during training.

To address this challenge, we introduce another structured variant of our framework, *Parallel-Unseen*. This model incorporates explicit inductive biases into the attention mechanism to enforce path isolation. Specifically, inspired by prior work (Yang et al., 2025b), we design a Multiverse model with corresponding customized attention masks and position ids to this goal.

### 3.4.1 Multiverse Attention Mechanism

We incorporate these inductive biases directly into the attention layer via Multiverse, as shown in Figure 2.

- Attention masks between different paths restricts tokens within a `<Path>` block to attend only to their own path and the shared context, preventing information leakage across sibling paths.

- Shared position encodings allocate each path a disjoint set of position indices, ensuring that parallel paths can begin decoding from identical positions without interference.

Together, these constraints enforce explicit isolation among reasoning paths while preserving visibility from the shared `<Summary>` block, which is essential for integrating insights across paths.

### 3.4.2 The Training Recipe and Reward Modeling

In preliminary experiments, we find that directly applying the progressive training curriculum from Parallel-R1-Seen to the structured variant proves ineffective. We attribute this to the poor generalization of attention masks from easy to hard problems (Yang et al., 2025c). To address this limitation, we remove the Stage 1 RL and redesign the reward schedule, and evaluate two alternative schemes.

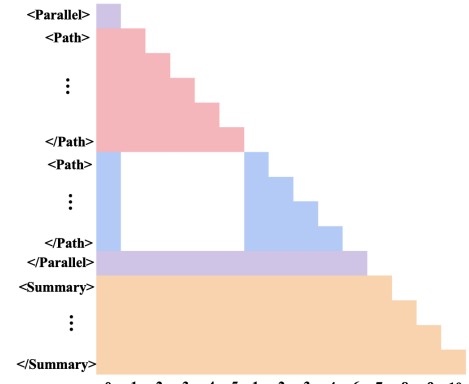

**(S1) Accuracy-only.** We optimize solely for correctness without encouraging parallel thinking.

**(S2) Alternating accuracy and parallel.** This scheme alternates between two different reward functions within fixed windows of W=10 steps. For 80% of the steps, we use a standard accuracy-only reward ($R_{\mathrm{acc}}$). For the remaining 20%, we use a tiered reward system to provide a nuanced incentive for parallel thinking: (i) +1.2: If the generated output contains

Figure 2: Illustration of the multiverse attention masks and position ids, where different tokens have distinct visible positions. Blank regions mean that tokens from separate paths cannot attend to each other.

at least one parallel thinking unit AND the final answer is correct; (ii) +1.0: If the generated output does not contain a parallel thinking unit AND the final answer is correct; (iii) -1.0: For all other cases with incorrect answers.

This schedule reintroduces a calibrated incentive for parallel thinking without allowing it to dominate training. Taken together, these reward designs enable *Parallel-R1-Unseen* to exploit the benefits of structural isolation while mitigating the risk of overfitting to superficial parallel patterns.

## 4 Experiments

### 4.1 Experimental Setups

We utilize Qwen-3-4B-Base (Yang et al., 2025a) as our backbone. Our implementation is adapted from the VERL codebase (Sheng et al., 2024), where we follow its official training recipe without any hyperparameter tuning. In the cold start stage, we perform SFT on our curated Parallel-GSM8K, using a batch size of 128, a learning rate of 1e-5, a weight decay of 0.01, and a warm-up step ratio of 0.1 with the cosine learning-rate schedule, resulting in 58/230 gradient update steps for Parallel-SFT-Seen and Parallel-SFT-Uneen, respectively. For Stage 1, we optionally perform RL on GSM8K for five epochs, using a batch size of 1024, 5 rollouts, and a learning rate of 1e-6 without warm-up or learning rate scheduling, resulting in 35 gradient update steps. For Stage 2, we perform RL on the DAPO training set for 300 gradient update steps, using a batch size of 512, a rollout of 8, and a learning rate of 1e-6 without warm-up or learning rate scheduling.

We measure our models on four standard mathematical reasoning benchmarks, including AIME'24, AIME'25, AMC'23, and MATH (Hendrycks et al., 2021). Since our training data (DAPO) consist solely of numeric answers, models are biased toward numeric outputs. To prevent evaluation artifacts

Table 2: Performance comparison on four mathematical reasoning benchmarks for the Qwen-3-4B-Base model trained under different sequential and parallel thinking configurations. For evaluation metrics, we report Mean@16 and Pass@16 for AIME25, AIME24, and AMC23, while MATH is assessed with Mean@1. # Parallel means the activation ratio of parallel thinking within all examples.

| Method | # Parallel | AIME25 | | AIME24 | | AMC23 | | MATH | Avg. |
|---|---|---|---|---|---|---|---|---|---|
| | | Mean@16 | Pass@16 | Mean@16 | Pass@16 | Mean@16 | Pass@16 | Mean@1 | |
| Qwen3-4B-Base | 0.0 | 5.5 | 24.6 | 10.0 | 25.4 | 39.3 | 74.6 | 54.0 | 27.2 |
| *SFT + Parallel* | | | | | | | | | |
| Parallel-SFT-Seen | 95.6 | 8.0 | 29.8 | 10.6 | 26.4 | 48.9 | 79.2 | 76.6 | 36.0 |
| Parallel-SFT-Unseen | 95.6 | 5.2 | 20.9 | 8.5 | 26.7 | 41.7 | 80.1 | 71.5 | 31.7 |
| *RL Approach* | | | | | | | | | |
| GRPO (DAPO) | 0.0 | 14.8 | 32.4 | 18.5 | 30.6 | 63.6 | 85.1 | 83.5 | 45.1 |
| + RL on GSM8K | 0.0 | 13.3 | 26.3 | 18.8 | 34.9 | 66.4 | 82.2 | 82.6 | 45.3 |
| Improved GRPO (DAPO) | 0.0 | 15.8 | 25.2 | 20.4 | 37.5 | 65.8 | 86.4 | 83.5 | 46.4 |
| Parallel-R1-Seen | 27.3 | **19.2** | 38.9 | **19.4** | 37.1 | **70.5** | 85.0 | **86.7** | **48.9** |
| Parallel-R1-Unseen (S1) | 13.6 | 17.7 | 37.8 | 18.3 | 33.2 | 69.7 | 88.9 | 82.6 | 47.1 |
| Parallel-R1-Unseen (S2) | **63.0** | 19.0 | **42.2** | 16.3 | 31.8 | 67.5 | **91.5** | 84.5 | 46.8 |

where correct LaTeX-formatted solutions in Math500 are judged as errors, we filter out LaTeX-containing problems. We set the sample temperature to 1.0 and max response length to 3000. On the MATH dataset, we generate one response per question. For the remaining three datasets, we sample 16 independent responses per question at the same temperature and report the average accuracy (i.e., mean@16) to reduce randomness, which is consistent with Wang et al. (2025c). We additionally report pass@16 to show the upper bound of our approach with both internal and external parallelism.

**Discussion of Training Resources and Requirements**   The Parallel-R1-Seen has the comparable training cost as the sequential baseline: ∼3.5 days on 8×40GB GPUs. The Parallel-R1-Unseen variant requires a longer runtime (∼6 days on the same hardware) because it must dynamically construct multiverse-style 4D attention masks at every RL rollout step. Unlike Multiverse, which preprocesses all 4D masks offline during supervised data construction, our RL setting must build these masks online for every trajectory, resulting in additional overhead.

## 4.2   MAIN RESULTS

Table 2 presents the results across four benchmarks: AIME25, AIME24, AMC23, and MATH. We compare our method against three baselines: 1) RL with GRPO algorithm directly on the DAPO training set, 2) RL with GRPO algorithm in two stages: first trained on the GSM8K data, then further trained with RL on the DAPO set. The second baseline is included to ensure fair comparison, and 3) SFT + RL: a sequential baseline that first performs supervised fine-tuning on the *Parallel-GSM8K* dataset (treating special tokens as plain text), followed by GRPO training on the DAPO dataset.

As shown in Table 2, our progressive Parallel-R1 framework proved to be the most effective approach, consistently outperforming all baseline methods. The top-performing autoregressive variant, `Parallel-R1-Seen`, achieved the highest average score of 48.9. This success stems from our curriculum learning strategy, which is designed to overcome the limitations of pure RL baselines. For example, while SFT provides a substantial initial improvement (i.e., 31.7 for `Parallel-SFT-Unseen` vs. 4.6 for the base model), it remains insufficient for advanced reasoning, as evidenced by its failure to reach the GRPO baseline score of 45.1. Furthermore, simply applying additional RL on easier data yields only marginal gains (i.e., 45.3 vs. 45.1) for the sequential baseline, further validating our cold-start strategy for targeted format and behavior learning. Additionally, the improved sequential baseline can outperform GRPO baseline by 1.1 points on average.

Our results also reveal several key design trade-offs. The superior performance of the autoregressive model compared to its multiverse counterparts suggests that explicit architectural modifications, while still leading to performance improvement over sequential baselines, can be detrimental to RL training. Second, the comparison between reward schedules in `Parallel-R1-Unseen (S1)` and `(S2)` demonstrates that reward design plays a pivotal role in balancing the parallel ratio against overall performance. A detailed discussion of these observations is provided in Section 4.3.3.

### 4.3 ANALYSIS

#### 4.3.1 ABLATION STUDY ON TRAINING APPROACH

In this section, we further explore the role of two-stage RL in our training pipeline. One natural question is whether learning on GSM8K, which is a relatively simple math dataset, truly benefits from the RL, given that the structural parallel reasoning format (e.g., the correct use of `<Parallel>`, `<Path>`, and `<Summary>` tokens) can also be acquired directly through SFT (Yang et al., 2025b).

Table 3: **Ablation Study on Training Approach:** comparison of different training configurations.

| Training Configuration | AIME25 | AIME24 | AMC23 | MATH | Avg. |
|---|---|---|---|---|---|
| **Effect of Training Stages** | | | | | |
| Parallel-R1-Seen | **19.2** | **19.4** | **70.5** | **86.7** | **48.9** |
|   - w/o RL on GSM8K | 17.9 | 19.0 | 65.0 | 84.5 | 46.6 |
| Parallel-R1-Unseen (S1) | 17.7 | 18.3 | 69.7 | 82.6 | 47.1 |
|   + with RL on GSM8K | 14.4 | 12.9 | 52.3 | 74.4 | 38.5 |
|   - SFT on GSM8K (# Parallel $\approx$ 0) | 15.0 | 18.8 | 62.8 | 83.9 | 45.1 |
| **Effect of Parallel Thinking Prompt** | | | | | |
| Parallel-R1-Seen | 19.2 | **19.4** | **70.5** | **86.7** | **48.9** |
|   - w/o Parallel Thinking Prompt | **20.4** | 16.5 | 66.7 | 84.8 | 47.1 |

Table 3 presents the results of this ablation study. For *Parallel-Seen*, removing the RL training on GSM8K in Stage 1 leads to a consistent performance drop (–2.3% on average). This indicates that learning format through SFT alone is insufficient. Without RL on easy tasks, the model enters Stage 2 training on more difficult general math without having acquired the ability to trigger or use parallel thinking adaptively. As a result, Stage 2 RL must both discover adaptive parallel-thinking behavior and strengthen mathematical reasoning on challenging tasks, which is more difficult to optimize. Additionally, the SFT stage provides the structural prior required for RL to explore parallel thinking. When we remove the Parallel-GSM8K SFT, parallel-thinking behaviors are rarely activated during RL, and the policy regresses to standard sequential generation.

Interestingly, the multiverse variant exhibits the opposite trend: adding Stage 1 RL on GSM8K significantly degrades performance (–8.6% on average). We hypothesize that this occurs because the structured attention mask learned on easy math tasks (GSM8K) does not transfer well to more complex problems, leading to a distribution shift and overfitting to superficial patterns, which is consistent with the findings in Yang et al. (2025c). This contrast highlights a key insight: while RL on easy math is crucial for the autoregressive variant to bootstrap adaptive parallel thinking, multiverse variants require a different training recipe and reward schedule to generalize effectively.

#### 4.3.2 ABLATION STUDIES ON PARALLEL THINKING PROMPT

We also conduct ablation studies on the effect of our parallel thinking prompt. As shown in Table 3, removing the prompt results in a 1.8% performance drop on average. It indicates that providing detailed instructions during training helps the model better understand the parallel thinking process.

#### 4.3.3 ABLATION STUDIES ON REWARD MODELING

We further answer one research question: how to effectively stimulate parallel thinking behavior? We test three reward modeling methods: direct accuracy, direct parallel, and an alternating approach. Table 4 shows the results. First, the "Accuracy" configuration, which optimizes solely for problem correctness, yields the highest average performance on two out of four benchmarks, particularly on the AMC dataset (69.7). However, this approach yields a very low parallel ratio of 13.6. In contrast, the "Parallel" configuration, which directly rewards the generation of parallel structures, achieves a high parallel ratio of 80.3. However, this focused optimization leads to a significant performance drop across most benchmarks. Our Alternating Acc./Parallel strategy provides a superior balance.

Table 4: **Ablation Study on Reward Modeling:** comparison of different training configurations.

| Training Configuration | Parallel Ratio | AIME 25 | AIME 24 | AMC 23 | MATH |
|---|---|---|---|---|---|
| Accuracy | 13.6 | 17.7 | **18.3** | **69.7** | 82.6 |
| Parallel | **80.3** | 17.7 | 15.2 | 59.4 | 81.7 |
| Alternating Acc./Parallel | 63.0 | **19.0** | 16.3 | 67.5 | **84.5** |

Table 5: Diversity analysis on parallel thinking variants.

| Method | Token-Level Diversity (BLEU ↓) | Semantic Diversity (Cosine ↓) |
|---|---|---|
| Parallel-SFT-Unseen | 0.0675 | 0.6360 |
| Parallel-R1-Unseen (S2) | 0.0627 | 0.6083 |

## 4.4 ANALYSIS ON DIVERSITY OF PARALLEL THINKING

In this section, we investigate the diversity of paths within each generated parallel thinking block. Specifically, we utilize BLEU and cosine similarity to measure token-level diversity and semantic diversity, respectively. For token-level diversity, we compute pairwise BLEU scores between all paths within the same parallel block using `sentence_bleu` function from the NLTK library. For semantic diversity, we encode each path using the widely adopted Sentence Transformer model `all-MiniLM-L6-v2`, which produces 384-dimensional sentence embeddings. We then compute pairwise cosine similarity between all encoded paths within each block. We report diversity results using the Parallel-SFT-Unseen and Parallel-R1-Unseen (S2) variants because they exhibit consistently high parallel ratios, ensuring that most examples contain multiple valid paths. This makes pairwise diversity computation well-defined and meaningful. We present the results in Table 5. The values indicate that, in both settings, the generated paths are not identical, and multiple distinct trajectories are present within each parallel block.

## 4.5 EVOLUTION OF PARALLEL THINKING BEHAVIOR DURING RL TRAINING

To gain deeper insight into how the model's strategy evolves during training, we analyzed the positional dynamics of the `<Parallel>` block in the RL process. We measured the relative position of each block by dividing its starting token index by the total sequence length of the solution. The training dynamics in Figure 3 showcase a trend: the average relative position of the `<Parallel>` block steadily increases as the RL training step ascends.

We hypothesize that this positional shift may reflect a more conservative, risk-averse strategy to maximize its reward, a behavior shaped directly by the final-answer-dominated reward design. In the early stages of training, when the model's reasoning ability is weak, using parallel paths for **computational exploration** is a necessary, high-variance strategy to discover a potential solution. However, as the model's core reasoning ability improves, such early-stage exploration becomes a liability that might introduce errors and jeopardize the final reward.

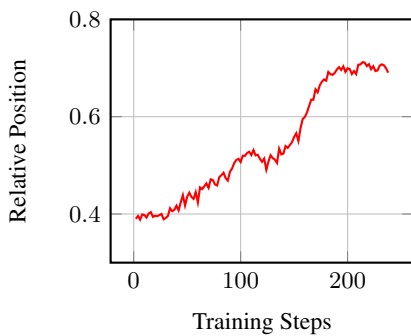

Figure 3: Dynamics of the relative position of the `<Parallel>` block during RL training. The increasing trend indicates the model learns to apply parallel thinking later in the reasoning process.

Consequently, the model learns a more risk-averse strategy to secure a correct answer. It first derives a solution using a single, high-confidence reasoning path. Only after a potential answer is found, it deploys the `<Parallel>` block for **multi-perspective verification**. This late-stage use of parallel thinking confirms the result without risking the integrity of the primary reasoning path, thus maximizing the probability of receiving a positive reward. This observed behavior is consistent with a potential trade-off between final-answer optimization and the preservation of diverse reasoning structures. Two case studies illustrating this evolution are further provided in Appendix M. Additional analyses in Appendix H show similar trends on a larger model and a different domain.

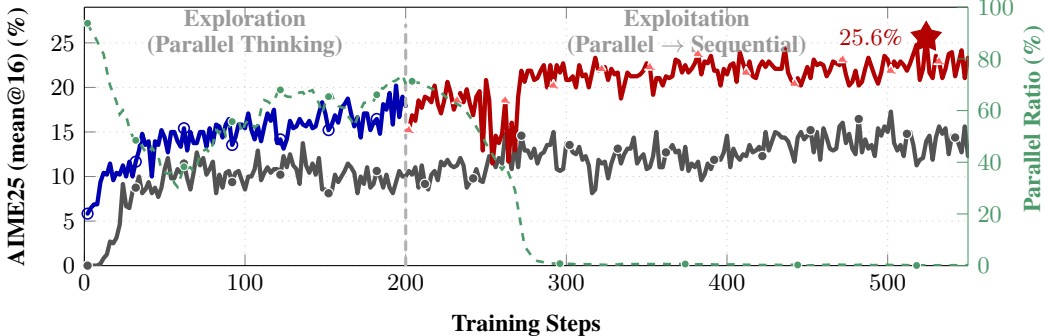

Figure 4: **Two-stage training with parallel reasoning as a mid-training exploration scaffold.** Left axis plots AIME25 accuracy for Baseline (gray), Stage-1 (blue), and Stage-2 (red); right axis shows the proportion of outputs using the explicit parallel thinking structure. Stage-1 (0–200 steps; vertical dashed line) alternates ACC/PAR rewards to promote exploration, while Stage-2 continues GRPO with an accuracy reward only and is plotted after a +200-step shift to align the timeline. As training transitions from parallel to sequential reasoning, the parallel ratio decreases yet accuracy continues to improve, peaking at 25.6%, which exceeds the sequential baseline trained via GRPO.

### 4.6 Extra Bonus: Parallel Thinking as a Mid-Training Exploration Strategy

This section explores whether parallel thinking can serve as a potential exploration scaffold to improve RL performance. To verify this, we design a two-stage training curriculum as follows.

- **Stage-1 (Exploration Phase, before 200 steps)**: The primary goal of this initial phase is to maximize exploration. In this stage, we follow the training approach of our Parallel-R1-Unseen (S2) As shown by the green dashed line in Figure 4, this scheduler ensures a much higher parallel ratio, encouraging the model to constantly explore a wide breadth of reasoning paths.
- **Stage-2 (Exploitation Phase, after 200 steps)**: At the 200-step mark, we change the focus from exploration to exploitation. The training objective is then switched to optimize for accuracy alone, allowing the model to exploit the effective strategies discovered during the exploration phase. We choose 200 steps as a switching point because, empirically, the parallel-structure behavior shows a noticeable transition around this region (Figure 3), though this choice is not theoretically optimal

As illustrated in Figure 4, upon entering stage 2, the model's performance (red line) improves, reaching a peak AIME25 accuracy of 25.6%, a notable 42.9% improvement over the Baseline GRPO model. Critically, this performance gain occurs even as the model's reliance on the parallel structure decreases (as shown by the declining parallel ratio in stage 2). This observation provides preliminary evidence that the benefit of parallel thinking may come not only from its explicit structure, but also from the broader policy space it encourages during exploration. In this sense, the initial forced exploration acted as a scaffold, guiding the model to a more effective region in the policy space, from which it could then learn a final policy.

## 5 Conclusion

In this work, we presented **Parallel-R1**, the **first reinforcement learning framework** to teach large language models to perform parallel thinking from scratch on real-world mathematical reasoning tasks. We proposed a **progressive training curriculum**, enabled by a simple data pipeline, that successfully bootstraps this complex skill by separating the learning of format, behavior, and core reasoning into distinct stages. Our approach achieved consistent accuracy improvements on several challenging mathematical reasoning benchmarks compared to strong baselines. Our analysis yielded several key insights into the learning dynamics. We discovered that the model learns a risk-averse strategy, shifting its use of parallel thinking from early-stage **computational exploration** to late-stage **multi-perspective verification**. Most significantly, we provide preliminary evidence that parallel thinking may function as a mid-training exploration scaffold, where a temporary forced-exploration phase appears to help the model reach a stronger final policy in our experiments.

ETHICS STATEMENT

Our research focuses on the development of reinforcement learning frameworks to teach LLMs adaptive parallel thinking ability. The goal is to advance more effective and interpretable reasoning systems that can solve complex mathematical problems. We conduct rigorous evaluation on publicly available benchmarks and ensure transparency in our training and analysis procedures. This work does not involve any ethical risks related to data privacy or misuse and is intended solely to promote research progress in reasoning with LLMs.

REPRODUCIBILITY STATEMENT

We have made significant efforts to ensure the reproducibility of Parallel-R1:

- **Training/Evaluation datasets:** The cold-start dataset Parallel-GSM8K, constructed from GSM8K with parallel-thinking prompting, will be released upon publication. All other dataset/benchmarks used (DAPO, AIME, AMC, MATH) are publicly available. Detailed information about data generation, filtering, and format validation is documented in Section 3.2 and Appendix L.
- **Code:** We have given enough details in Section 3, Section 4.1, Appendix D to reproduce our results and we will open-source the code as soon as we collect our scripts for an easy way to reproduce our results.
- **Experimental Configuration:** We provide detailed experimental specifications, hyperparameter configurations, and procedural details are documented in Section 4.1.

By releasing these resources, we aim to facilitate transparent comparison, ensure reproducibility, and encourage further research on reinforcement learning for parallel reasoning.

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

## A    THE USE OF LLMS

We only use LLMs to polish the paper writing.

## B    LIMITATIONS AND FUTURE DIRECTION

To the best of our knowledge, this work provides the first systematic investigation of learning explicit parallel-thinking behaviors via reinforcement learning on general mathematical reasoning tasks. Because parallel thinking is a novel and non-native reasoning strategy for current LLMs, we adopt a cold-start SFT stage that introduces primarily the structural format of parallel thinking, avoiding the sophisticated teacher-constructed reasoning patterns used in Multiverse (Yang et al., 2025b). We also start from a base model to avoid inheriting strong sequential reasoning priors, which could otherwise interfere with the learning of parallel thinking during RL. This controlled setup enables a clean examination of RL-driven behavioral emergence without confounding factors

A complementary line of research is to study how strong sequential reasoning model can be transformed into parallel thinking models without sacrificing performance. This is an interesting and orthogonal direction, but falls outside the scope of our investigation, which centers on understanding RL-driven behavioral emergence. Future work may explore extending Parallel-R1 to larger backbones, stronger reasoning foundations (e.g., reasoning-enhanced SFT or Multiverse-style data), and broader domains such as open-ended creative writing, multi-modal reasoning and critic models (Xiong et al., 2025b; 2026).

## C    BACKGROUNDS OF GRPO (GUO ET AL., 2025)

Let $q$ be a question, and let $\{o_i\}_{i=1}^{G}$ be $G$ candidate responses sampled from the old policy $\pi_{\theta_{\text{old}}}(\cdot \mid q)$. We denote $r_i$ as the reward for $o_i$. We define:

$$\rho_i = \frac{\pi_\theta(o_i \mid q)}{\pi_{\theta_{\text{old}}}(o_i \mid q)}, \qquad \overline{r} = \frac{1}{G}\sum_{j=1}^{G} r_j, \qquad A_i = \frac{r_i - \overline{r}}{\sqrt{\frac{1}{G}\sum_{j=1}^{G}(r_j - \overline{r})^2 + \varepsilon_{\text{stab}}}},$$

where $\varepsilon_{\text{stab}}$ is a constant for numerical stability and $A_i$ is the advantage. The GRPO loss is then:

$$\mathcal{L}_{\text{GRPO}}(\theta) = \mathbb{E}_{\substack{q\sim\mathcal{D} \\ \{o_i\}\sim\pi_{\theta_{\text{old}}}}} \left[ \frac{1}{G}\sum_{i=1}^{G} \min\!\big(\rho_i A_i,\ \text{clip}(\rho_i, 1-\alpha, 1+\alpha)\, A_i\big) \ - \ \beta\, D_{\text{KL}}\big(\pi_\theta \,\|\, \pi_{\text{ref}}\big) \right].$$

## D    PROMPTS

> **Baseline Prompt**
>
> ```
> {Problem} Let's think step by step and output the final answer
>     after "Final Answer: ".
> ```

> **Parallel Thinking Prompt**
>
> ```
> Solve the following problem step by step.
> During the reasoning process, whenever you encounter a step that
>     may benefit from multiple perspectives or independent reasoning
>     , insert a <Parallel> block at that point.
>
> Within each <Parallel> block:
>
> Include at least two distinct and independent reasoning paths.
> ```

```
Each path must be enclosed within <Path> and </Path> tags.
Do not include any ordering information or cross-references between
    paths, as they are generated simultaneously and independently.
Close the block with </Parallel>.
Immediately after each </Parallel>, write a concise summary of
    insights or conclusions drawn from all paths, enclosed in <
    Summary> and </Summary> tags.

Repeat this process adaptively as needed throughout the reasoning.
Do not explicitly mention that you are triggering parallel thinking
    -just insert the <Parallel> block naturally within the
    reasoning chain.

End your response with a line starting with Final Answer: followed
    by the final result.

Problem: {Problem}
```

## E   EXAMPLES OF *Parallel-GSM8K*

### Example of *Parallel-GSM8K*

```
Dulce's father has eight orange trees, each with 200 fruits. Dulce
    picks 2/5 of the oranges from each tree, so we need to find the
     total number of fruits remaining.

First, let's find out how many fruits are picked from one tree. We
    can compute this using different strategies.

<Parallel>
<Path>
Compute 2/5 of 200 by multiplying the fraction and the number:
    (2/5) * 200 = (2 * 200) / 5 = 400 / 5 = 80
</Path>
<Path>
Compute 2/5 of 200 by dividing the number by the denominator first:
     200 Ã· 5 = 40, then multiply by the numerator: 40 * 2 = 80
</Path>
</Parallel>
<Summary>The parallel block showed two methods to calculate the
    number of fruits picked from one tree, both resulting in 80
    fruits.
</Summary>

With 80 fruits picked from one tree, the number of fruits remaining
     in one tree can be found by subtracting or directly using the
    fraction.

<Parallel>
<Path>
Calculate the remaining fruits by subtracting the picked amount
    from the total: 200 - 80 = 120
</Path>
<Path>
Calculate the remaining fruits directly using the fraction: (3/5) *
     200 = (3 * 200) / 5 = 600 / 5 = 120
</Path>
</Parallel>
```

```
<Summary>
The parallel block showed two ways to determine the fruits
    remaining per tree, confirming 120 fruits left in each tree.
</Summary>

Since there are eight trees, the total number of fruits remaining
    is found by multiplying the per-tree remaining amount by the
    number of trees.

120 * 8 = 960

Final Answer: 960
```

Table 6: StrategyQA accuracy comparison.

| Method | Accuracy (%) | Δ over Baseline |
|---|---|---|
| GPRO (Baseline) | 85.7 | – |
| Parallel-R1-Seen | **89.7** | **+4.0** |
| Parallel-R1-Unseen (S1) | 86.8 | +1.1 |
| Parallel-R1-Unseen (S2) | 86.6 | +0.9 |

Table 7: Ablation on structural reward weight, window-size, and frequency settings.

| Setting | AIME24 | AIME25 | AMC23 | MATH |
|---|---|---|---|---|
| Weight = 1.2 (Parallel-R1-Unseen (S2)) | 16.3 | 19.0 | 67.5 | 84.5 |
| Weight = 2.0 | 17.7 | 17.9 | 64.5 | 84.2 |
| Window-size = 10 (Parallel-R1-Unseen (S2)) | 16.3 | 19.0 | 67.5 | 84.5 |
| Window-size = 20 | 17.5 | 19.4 | 65.3 | 84.2 |
| Frequency = 8/2 (Parallel-R1-Unseen (S2)) | 16.3 | 19.0 | 67.5 | 84.5 |
| Frequency = 6/4 | 18.2 | 19.6 | 65.6 | 83.9 |

# F  GENERALIZATION TO NON-MATH REASONING TASKS

In this section, we provide preliminary investigation on whether Parallel Thinking can be generalized to domains beyond mathematical reasoning tasks. To answer this question, we evaluated on the StrategyQA task. We present the results in Table 6. Despite being trained exclusively on mathematical reasoning, the Parallel-R1 variants still outperform the sequential RL baseline on StrategyQA. This preliminary evidence reveals potential for extending Parallel Thinking to broader tasks.

# G  SENSITIVE ANALYSIS ON REWARD MODELING

We provide sensitive analysis on rewarding modeling. We tested several variations of the reward schedule on Parallel-R1-Unseen (S2), including changes to

- alternating frequency (e.g., 8/2, 6/4),
- parallel weighting (e.g., 2.0, 1.2), and
- alternating window-size (10, 20).

We can see our Parallel-R1-Unsee (S2) is robust across these weight, window size and frequency choices.

# H  ROBUSTNESS ANALYSIS ON OBSERVATION IN SECTION 4.5

In this section, we provide additional robustness analyses to support the observation in Section 4.5, namely the shift of the parallel thinking behaviors from early-stage computation to late-stage verification during RL training. We report results on a larger backbone model, Parallel-R1-Unseen (S2) with Qwen2.5-7B-Math, and on a different domain (StrategyQA) in Table 8 and Table 9, respectively. These results show that the same monotonic upward trend consistently appears across both a stronger backbone and a non-mathematical domain, confirming that the behavioral evolution is not a training artifact but a stable and model-agnostic pattern.

# I  FURTHER ANALYSIS ON PERFORMANCE IMPROVEMENT VARIANCE BETWEEN AIME24 AND AIME25.

In Table 2, we observe that the pass@16 improvement on AIME24 is smaller than that on AIME25. To better understand this discrepancy, we conduct a more detailed analysis across the two datasets.

Table 8: Mean relative position of the `<Parallel>` block at selected RL steps for Parallel-R1-Unseen (S2) (Qwen2.5-7B-Math).

| Model | 20 | 40 | 60 | 80 | 100 | 120 |
|---|---|---|---|---|---|---|
| Parallel-R1-Unseen (S2) | 0.4652 | 0.5269 | 0.5707 | 0.5964 | 0.6364 | 0.6508 |

Table 9: Mean relative position of the `<Parallel>` block on StrategyQA at early vs. late RL steps.

| Dataset | 20 | 200 |
|---|---|---|
| StrategyQA | 0.3875 | 0.5464 |

Our hypothesis is that AIME25 contains a higher proportion of problems whose structure or reasoning patterns benefit more from parallel thinking.

**Cross-Dataset Comparison of Problem Demand for Parallel Thinking.** To empirically test this hypothesis, we perform a cross-dataset comparison using an external LLM judge (GPT-4o-mini). For every pair of problems $(A_i, B_j)$ where $A_i$ comes from AIME24 and $B_j$ comes from AIME25, the judge is asked to determine which problem is more likely to benefit from explicit parallel thinking. In total, this yields $30 \times 30 = 900$ pairwise comparisons.

To verify reliability, we manually inspect 50 randomly sampled pairs and find the LLM judgments highly consistent with human evaluation. The aggregated results, summarized in Table 10, show that AIME25 problems are judged to have a stronger demand for parallel thinking. This observation aligns with the larger improvement observed on AIME25 and suggests that the variance in gains is primarily driven by inherent differences in problem structure rather than instability of the method.

Table 10: Pairwise comparison of problem-level demand for parallel thinking between AIME24 and AIME25. Results are based on 900 LLM-judged pairwise comparisons.

| Dataset | Wins | Losses | Total | Win Rate |
|---|---|---|---|---|
| AIME25 | **686** | 214 | 900 | **0.762** |
| AIME24 | 214 | **686** | 900 | 0.238 |

## J  ADDITIONAL COMPARISON WITH MULTIVERSE

The original Multiverse model from Yang et al. (2025b) is based on Qwen-2.5-32B-Instruct, making it difficult to directly compare with our Parallel-R1 framework, which is built upon the Qwen-3-4B-Base model. To ensure a fair comparison, we re-implemented a *Multiverse-SFT-4B* model by training Qwen-3-4B-Base using the Multiverse parallel-thinking data and prompt format. This yields a size-matched Multiverse variant under comparable training conditions.

The results are summarized below. We observe that all Parallel-R1 variants outperform the Multiverse-SFT-4B baseline across AIME24/25, AMC23, and MATH. This supports our central finding that *RL-induced parallel thinking* can achieve stronger reasoning performance than *teacher-induced parallel thinking* derived from rewriting pipelines.

## K  ADDITIONAL COMPARISON WITH STRONGER TEST-TIME PARALLEL THINKING BASELINE

We include Self-Consistency@16 (SC@16), which is a widely used test-time parallel reasoning baseline. We report the results below. We observe that: 1) The sequential RL baseline with SC@16 achieves performance that is comparable to or lower than our Parallel-R1 variants across most bench-

Table 11: Comparison of performance with Multiverse-4B across AIME24, AIME25, AMC23, and MATH.

| Method | AIME24 | AIME25 | AMC23 | MATH |
|---|---|---|---|---|
| Multiverse-SFT-4B | 8.2 | 10.6 | 49.0 | 76.8 |
| GRPO (DAPO) | 18.5 | 14.8 | 63.6 | 83.5 |
| Parallel-R1-Seen | 19.4 | 19.2 | 70.5 | 86.7 |
| Parallel-R1-Unseen (S1) | 18.3 | 17.7 | 69.7 | 82.6 |
| Parallel-R1-Unseen (S2) | 16.3 | 19.0 | 67.5 | 84.5 |

Table 12: Comparison of performance with test-time parallel thinking baselines across AIME25, AIME24, AMC23, and MATH.

| Method | AIME25 | AIME24 | AMC | MATH |
|---|---|---|---|---|
| GRPO (DAPO) | 14.8 | 18.5 | 63.6 | 83.5 |
| GRPO (DAPO) + SC@16 | 18.9 | 19.0 | 65.3 | 84.8 |
| Parallel-R1-Seen | 19.2 | 19.4 | 70.5 | 86.7 |
| Parallel-R1-Unseen (S1) | 17.7 | 18.3 | 69.7 | 82.6 |
| Parallel-R1-Unseen (S2) | 19.0 | 16.3 | 67.5 | 84.5 |

marks; and 2) Although SC@16 can further improve the baseline performance, it requires nearly 16× more inference tokens, which is not efficient in token usage.

## L  PARALLEL THINKING FORMAT CHECK

---

**Algorithm 1** Parallel Thinking Format Check

---

**Input:** tokens – list of tokens from the parallel-thinking trace;
      tag_pairs – set of valid (opening, closing) tag pairs, e.g. $\{(\texttt{<Path>}...\texttt{</Path>}), ...\}$
**Output:** format_valid – boolean indicating whether the trace is well-formed

1:   $S \leftarrow \emptyset$
2:   $format\_valid \leftarrow$ true
3:   **for all** $t$ **in** tokens **do**
4:      **if** $t$ **is an opening tag then**
5:         push $t$ onto $S$
6:      **else if** $t$ **is a closing tag then**
7:         **if** $S$ is empty **then**
8:            $format\_valid \leftarrow$ false
9:            **break**
10:         **end if**
11:         $top\_tag \leftarrow \text{Top}(S)$
12:         **if** $(top\_tag, t) \in$ tag_pairs **then**
13:            pop $S$
14:         **else**
15:            $format\_valid \leftarrow$ false
16:            **break**
17:         **end if**
18:      **end if**
19: **end for**
20: **if** $format\_valid$ **and** $S \neq \emptyset$ **then**
21:     $format\_valid \leftarrow$ false
22: **end if**
23: **return** $format\_valid$

---

## M  CASE STUDIES: EXPLORATION VS. VERIFICATION

We present two cases illustrating the evolution of parallel thinking: an early-stage model using it for exploration, and a late-stage model applying it for verification.

---

**Example of Parallel Exploration**

```
Case Study: Find the sum of all integer bases $b>9$ for which
    $17_b$ is a divisor of $97_b$.

  First, the number $17_b$ in base $b$ is equal to $1 \times b + 7
      = b + 7$ in base 10.
  Similarly, $97_b$ is equal to $9 \times b + 7 = 9b + 7$ in base
      10.
  We need to find all integer bases $b > 9$ such that $b + 7$
      divides $9b + 7$.

  At this point, consider two distinct strategies for finding $b$
      :\\

<Parallel>
<Path>
Factor theorem approach: Consider 9b + 7 as a polynomial in b and
    use the factor theorem.
Since b + 7 divides 9b + 7, by the factor theorem, 9(-7) + 7 = -63
    + 7 = -56 must be divisible by b + 7.
Since b > 9, b + 7 > 16, and -56 can be factored as -56x1, -28x2,
    -14x4, or -7x8 (and their positive counterparts).
Only b + 7 = 14 gives b = 7, but b > 9, so no solution from this
    path.
However, we should also consider the positive factors for
    completeness.
The positive factorization 56x1, 28x2, 14x4, or 7x8 yields b + 7 =
    56 (giving b = 49),
b + 7 = 28 (giving b = 21), b + 7 = 14 (giving b = 7, invalid),
and b + 7 = 8 (giving b = 1, invalid).
Thus, the only valid solution from this path is b = 21 and b = 49.
</Path>
<Path>
Using base arithmetic: In base b, 97\_b - 1x17\_b = 97\_b - 17\_b =
     80\_b,
which is 8xb + 0 = 8b in base 10.
Since 9b + 7 = 8b + (b + 7), b + 7 must divide 8b, implying that
    for some integer k,
9b + 7 = (b + 7)(8 + k).
Simplifying, 9b + 7 = 8b + 8k + kb + 7k, so b(1 - k) = 8k + 7.
Since b > 9, 1 - k must be positive or negative to keep b an
    integer.
If 1 - k = 1, then b = 8k + 7 with k < 1.
Possible integer k values are k <= 0, but k = 0 gives b = 7 <= 9,
and k < 0 gives negative b, which is invalid.
Thus, this path leads to no valid solutions directly.
However, it reinforces that b + 7 must divide 8b.
</Path>
</Parallel>
<Summary>
These two approaches consider both the algebraic factorization
    method
and an alternative numerical or arithmetic approach to find
    possible values of b,
```

```
but only the factorization method yields valid bases greater than
    9.
</Summary>

Next, consider the condition that $b + 7$ divides $9b + 7$.
From the earlier, $9b + 7 = 9(b+7) - 56$.
For $b + 7$ to divide $9b + 7$, it must divide $56$.
Since $b + 7 \ge 17$ for $b > 9$, the only divisors of 56 exceeding
    16 are 28 and 56, giving $b+7\in\{28,56\}$ and thus $b\in
    \{21,49\}$.

Finally, verify:

For $b = 21$: $17_{21} = 21 + 7 = 28$, $97_{21} = 9 \times 21 + 7 =
    196$, $196 \div 28 = 7$.\\
For $b = 49$: $17_{49} = 49 + 7 = 56$, $97_{49} = 9 \times 49 + 7 =
    448$, $448 \div 56 = 8$.\\[4pt]
Thus, the sum is $21 + 49 = 70$.
Final Answer: 70
```

## Example of Multi-Perspective Verification

```
Case Study: Find the sum of all integer bases $b>9$ for which
    $17_b$ is a divisor of $97_b$.

First, I need to find all integer bases $b > 9$ for which $17_b$ is
     a divisor of $97_b$. To do this, I'll start by converting
    these base-$b$ numbers to base 10, as that's more familiar.

In base $b$, the number $17_b$ means $1 \times b + 7 = b + 7$ in
    base 10. Similarly, $97_b$ means $9 \times b + 7 = 9b + 7$ in
    base 10.

So, the condition that $17_b$ divides $97_b$ translates to $b + 7$
    dividing $9b + 7$ in base 10.

Now, for $b + 7$ to divide $9b + 7$, the remainder when $9b + 7$ is
     divided by $b + 7$ must be zero. I can perform polynomial
    division or use the remainder theorem to find this.

Let me compute the remainder of $9b + 7$ divided by $b + 7$.

Divide $9b + 7$ by $b + 7$:
- The quotient is 9, since $9 \times (b + 7) = 9b + 63$.
- Subtract this from $9b + 7$: $(9b + 7) - (9b + 63) = 9b + 7 - 9b
    - 63 = -56$.

So, the remainder is $-56$. For $b + 7$ to divide $9b + 7$, it must
     divide $-56$, or equivalently, $b + 7$ must be a positive
    divisor of 56.

The positive divisors of 56 are: 1, 2, 4, 7, 8, 14, 28, 56.

Since $b > 9$, we have $b + 7 > 16$. So, $b + 7$ must be a divisor
    greater than 16.
- $b + 7 = 28 \Rightarrow b = 21$ (valid)
- $b + 7 = 56 \Rightarrow b = 49$ (valid)

So, the valid bases are $b = 21$ and $b = 49$. Now, I need to find
    the sum: $21 + 49 = 70$.
```

```
But before finalizing, I should confirm that these are the only
    valid bases. From the divisor check, only $b + 7 = 28$ and $b +
     7 = 56$ give $b > 9$, so no others.

<Parallel>
<Path>One might consider that $ b + 7 $ not only divides 56 but
    also that it must be such that $ 17_b $ and $ 97_b $ are valid
    base-$b$ numbers, which requires $ b > 7 $ (since digits 9 and
    7 are used). However, since $ b > 9 $ is given, this is already
     satisfied, and the divisor condition is the key.</Path>
<Path>Use the fact that for $ b + 7 $ to divide $ 9b + 7 $, the
    remainder must be zero, so we can solve $ 9b + 7 = k(b + 7) $
    for integer $ k $, but it's simpler to compute the remainder
    directly as above.</Path>
</Parallel>
<Summary>
The parallel block explored alternative strategies for solving $ b
    + 7 $ divides $ 9b + 7 $, including finding divisors,
    considering base validity, and solving equations, but
    ultimately confirmed that only $ b = 21 $ and $ b = 49 $
    satisfy the condition for $ b > 9 $.
</Summary>

Now, sum the valid bases: $21 + 49 = 70$.
Final Answer: 70
```

