# OpenReview forum: "Parallel-R1: Towards Parallel Thinking via Reinforcement Learning"
_ICLR.cc/2026/Conference — ICLR 2026 Poster_

### Official Review · Reviewer_6K5f · 2025-10-20

**Soundness:** 3
**Presentation:** 3
**Contribution:** 3
**Rating:** 8
**Confidence:** 3

**Summary:**

The paper introduces Parallel-R1, the first reinforcement learning (RL) framework that successfully trains large language models (LLMs) to perform parallel thinking—i.e., exploring multiple reasoning paths simultaneously—on complex real-world tasks like advanced math problems.

**Strengths:**

The paper identifies a critical gap: while test-time parallel reasoning (e.g., self-consistency, tree-of-thoughts) is common, models do not intrinsically learn this capability through training. By reframing parallel thinking as a trainable skill—not just an inference-time trick—the work addresses a foundational limitation in LLM reasoning. This aligns with cognitive science (humans use parallel exploration) and practical needs (e.g., robustness in math/STEM tasks), giving the research strong motivation and applicability.

The paper doesn’t just apply RL—it carefully interrogates how to reward parallel thinking. The alternating reward scheme (switching between accuracy-only and accuracy+parallelism rewards) demonstrates deep understanding of RL pitfalls:
1. Pure accuracy rewards suppress parallelism;
2. Pure structural rewards encourage “empty” parallelism without reasoning gains.
Extensive ablations (e.g., removing Stage 1 RL, testing prompts, comparing reward schemes) provide strong causal evidence for design choices, enhancing scientific credibility.

**Weaknesses:**

The paper operationalizes parallel thinking via syntactic control tags (<Parallel>, <Path>, <Summary>), but does not verify whether the content within <Path> blocks is meaningfully diverse or merely superficially duplicated reasoning. High “parallel ratio” (e.g., 95.6% in SFT models) could reflect template compliance without genuine exploration.

Evidence: Table 2 shows SFT models achieve near-perfect parallel activation but underperform RL variants (e.g., Parallel-SFT-Seen: 36.0 avg vs. Parallel-R1-Seen: 48.9). This suggests format adherence ≠ functional parallelism. Yet the paper lacks metrics for semantic diversity across paths (e.g., path divergence, mutual information, or solution disagreement rates).

All experiments are confined to math reasoning benchmarks (GSM8K, MATH, AIME, AMC). While math is a natural domain for structured parallel exploration, the paper claims broader relevance to “complex real-world reasoning tasks” (Abstract, Sec 1). However, no evidence is provided for applicability to domains like commonsense reasoning, planning, or open-ended QA—where “critical steps” are less well-defined and verification is harder.

 While the paper claims parallel thinking leverages “negligible cost” via GPU parallelism (Sec 1), it reports no inference-time metrics: latency, memory usage, or FLOPs. In practice, generating multiple paths and summaries increases compute. Without this, the claim of practicality is unsubstantiated.

**Questions:**

see weakness

---

> ### Author Response · Authors · 2025-11-27
> **Author Response (1/2)**
>
> Thanks for your detailed reviews and recognizing the contribution of our work. We've incorporated your advice to revise our PDFs, modifying statements that might cause confusion.
>
> > The paper operationalizes parallel thinking via syntactic control tags ([object Object], [object Object], [object Object]), but does not verify whether the content within [object Object] blocks is meaningfully diverse or merely superficially duplicated reasoning. High “parallel ratio” (e.g., 95.6% in SFT models) could reflect template compliance without genuine exploration. Evidence: Table 2 shows SFT models achieve near-perfect parallel activation but underperform RL variants (e.g., Parallel-SFT-Seen: 36.0 avg vs. Parallel-R1-Seen: 48.9). This suggests format adherence ≠ functional parallelism. Yet the paper lacks metrics for semantic diversity across paths (e.g., path divergence, mutual information, or solution disagreement rates).
>
> We thank the reviewer for raising the crucial concern regarding whether a high parallel ratio translates to functional parallelism. We agree that format compliance alone is insufficient.
> To rigorously investigate this, we introduced two metrics: 1) Token-Level Diversity (BLEU) and 2) Semantic Diversity (Cosine Similarity), to measure the surface and semantic difference across parallel paths. We focused on our Unseen variant, which exhibits a high parallel ratio for all benchmarks, because it is the ideal setting to address the reviewer's concern about high parallel ratio. The results are shown below, where both the SFT model and our Parallel-R1 possess high diversity in their parallel thinking blocks.
>
> **Why SFT cannot work:** The SFT stage, serving as a cold-start, primarily teaches the syntactic format for parallel thinking. This results in spurious parallelism on unseen data. The model applies the structure rigidly but fails to identify when and where a logically valid branch is needed. Eventually, SFT models achieve low performance. By contrast, the RL stage successfully binds the parallel thinking behavior to task correctness using reward signals, guiding the model to utilize the structure strategically for functionally effective exploration and achieving higher accuracy.
>
>
> | Method                 | Token-Level Diversity (BLEU ↓) | Semantic Diversity (Cosine ↓) |  Parallel Ratio |  Acc. |
> |------------------------|--------------------------------|--------------------------------|--------------------------------|--------------------------------|
> | Parallel-SFT-Unseen    |             0.0675                   |            0.6360                    |             95.6                   |            31.7                    |
> | Parallel-R1-Unseen (S2)   |            0.0627                   |           0.6083                    |            63.0                    |            46.8                    |
>
>
> > All experiments are confined to math reasoning benchmarks (GSM8K, MATH, AIME, AMC). While math is a natural domain for structured parallel exploration, the paper claims broader relevance to “complex real-world reasoning tasks” (Abstract, Sec 1). However, no evidence is provided for applicability to domains like commonsense reasoning, planning, or open-ended QA—where “critical steps” are less well-defined and verification is harder.
>
> We acknowledge we currently mainly focus on math reasoning benchmarks as it is a natural domain that provides well-defined steps and clear reward signals for validating the effectiveness of  our approach. To address address your concern, we have taken the following actions:
> - Changing the statement from “complex real-world reasoning tasks” to “To address this issue, we propose Parallel-R1, the first reinforcement learning (RL) framework that instills parallel thinking for complex real-world reasoning tasks, e.g., mathematical reasoning tasks, beyond synthetic tasks.”
> - Adding small-scale generalization experiments on StrategyQA to show the applicability of our parallel thinking approach. (Results are presented below.)
>
>
> | Method             | StrategyQA Accuracy |
> |--------------------|---------------------------|
> | GPRO (Baseline)    |        85.7          |
> | Parallel-R1-Seen   |         89.7 (+4.0)  |
> | Parallel-R1-Unseen (S1) |       86.8 (+1.1)            |
> | Parallel-R1-Unseen (S2) |       86.6 (+0.9)            |

---

> > ### Author Response · Authors · 2025-11-27
> > **Author Response (2/2)**
> >
> > > While the paper claims parallel thinking leverages “negligible cost” via GPU parallelism (Sec 1), it reports no inference-time metrics: latency, memory usage, or FLOPs. In practice, generating multiple paths and summaries increases compute. Without this, the claim of practicality is unsubstantiated.
> >
> > We thank the reviewer for highlighting the ambiguity regarding the computational cost. We agree that the claim of "negligible cost" was unsubstantiated without systematic metrics. We wish to clarify that we originally want  to use this sentence as theoretical motivation for the one advantage of parallel thinking, instead of  a primary experimental finding. To avoid misunderstanding, we will revise the sentence as follows:
> >
> > Moreover, parallel thinking provides such benefits by leveraging the concurrent execution capability of modern GPUs to explore multiple reasoning paths in parallel [1].
> >
> >
> > This would not impact our core contributions, 1) providing the first RL framework to instill parallel thinking behaviors beyond synthetic tasks; 2) investigate key problems in RL, e.g., cold-start problem and reward modelling, model architecture effect, providing insights to future work; and 3) providing in-depth analysis about parallel thinking behaviors and reveal some empirical evidence on the function of parallel thinking.

---

### Official Review · Reviewer_RoLa · 2025-10-31

**Soundness:** 3
**Presentation:** 3
**Contribution:** 3
**Rating:** 6
**Confidence:** 3

**Summary:**

This paper introduces Parallel-R1, the first framework that uses reinforcement learning to internalize parallel thinking capabilities in large language models, enabling them to adaptively explore multiple reasoning paths simultaneously during inference.
The framework employs a progressive curriculum learning approach, starting with supervised fine-tuning to learn the basic format, followed by reinforcement learning to stabilize and generalize this ability across tasks of increasing difficulty.
Experiments demonstrate that Parallel-R1 significantly improves performance on multiple mathematical reasoning benchmarks and reveals the dynamic shift in parallel thinking during training—from an exploration tool to a verification mechanism.
What's more, the paper identifies parallel thinking as a mid-training exploration scaffold, whose exploratory effects guide the model to a more optimal policy space, enabling it to achieve final performance far exceeding baselines even after the scaffold is removed.

**Strengths:**

1. The problem is novel and significant. The paper targets a highly cutting-edge and important problem which is how to internalize the advanced reasoning capability of "parallel thinking" in large language models through training (rather than test-time techniques).
2. The method is systematic, comprehensive, and insightful. Instead of proposing an isolated algorithm, the paper presents a complete, end-to-end solution framework.
3. The experiments are thorough, and the depth of analysis exceeds conventional standards. The empirical analysis goes beyond superficial "performance improvement" to delve deeply into the underlying mechanisms.

**Weaknesses:**

1. Potentially insufficient comparison with the strongest baselines. The paper's primary baselines are sequential-thinking RL methods. However, in the field of parallel reasoning, there exist other powerful test-time techniques, such as various variants of the "Tree of Thoughts" approach.
2. Limited scope in validating generalization ability. All experiments in the paper are focused on the domain of mathematical reasoning. While mathematics serves as a touchstone for evaluating reasoning capabilities, this narrow focus restricts the support for the paper's claim of applicability to "general reasoning tasks."

**Questions:**

1. The study would benefit from a formal analysis of computational efficiency during training.
2. It is recommended to incorporate comparative analyses with other parallel thinking algorithms
3. The current evidence cannot distinguish whether the acquired "parallel thinking" is a general-purpose reasoning strategy or a specialized skill optimized for mathematical problems.

---

> ### Author Response · Authors · 2025-12-02
> **Author Response**
>
> We thank the reviewer RoLa for the constructive and insightful feedback. Below we address all concerns point by point.
>
> > W1&Q2: Potentially insufficient comparison with the strongest baselines. The paper's primary baselines are sequential-thinking RL methods. However, in the field of parallel reasoning, there exist other powerful test-time techniques, such as various variants of the "Tree of Thoughts" approach.
>
> Thank you for the suggestion. Tree-of-Thoughts (ToT) is indeed a strong and influential test-time parallel reasoning paradigm. We appreciate the reviewer highlighting this direction.
> A fair ToT evaluation requires careful tuning of prompting schemes, step-level verification, and search hyperparameters, and performance is known to be highly sensitive to these choices. Due to the limited rebuttal period, running a meaningful ToT comparison is unfortunately not feasible. Instead, we include Self-Consistency@16 (SC@16), which is a widely used test-time parallel reasoning baseline. We report the results below.
>
> We observe that: 1)  The sequential RL baseline with SC@16 achieves performance that is comparable to or lower than our Parallel-R1 variants across most benchmarks; and  2) Although SC@16 can further improve the baseline performance, it requires nearly ~16× more inference tokens, which is not efficient in token usage.
> ### Table: Comparison of Self-Consistency@16 and Parallel-R1 Variants
> | Method                               | AIME25 (%) | AIME24 (%) | AMC23 (%) |Math (%) |
> |--------------------------------------|------------|------------|-----------|------------|
> | GRPO (DAPO)                  |     14.8        |      18.5       |     63.6     |  83.5 |
> | GRPO (DAPO) + SC@16          |     18.9       |    19.0        |     65.3    | 84.8 |
> | Parallel-R1-Seen                     |        19.2    |      19.4      |      70.5     | 86.7  |
> | Parallel-R1-Unseen (S1)              |        17.7     |       18.3     |      69.7    |  82.6 |
> | Parallel-R1-Unseen (S2)              |      19.0      |       16.3     |      67.5    |  84.5 |
>
> >W2&Q3:  Limited scope in validating generalization ability. All experiments in the paper are focused on the domain of mathematical reasoning. While mathematics serves as a touchstone for evaluating reasoning capabilities, this narrow focus restricts the support for the paper's claim of applicability to "general reasoning tasks."
> Thanks for the suggestion. We have additionally evaluated our approach on StrategyQA to examine whether the induced parallel-thinking behavior can transfer beyond mathematical settings. The results are shown below.
> Meanwhile, to better reflect the actual scope of our experiments, we have revised the claim from “general reasoning tasks” to “general mathematical reasoning tasks.”
>
>
> | Method             | StrategyQA Accuracy |
> |--------------------|---------------------------|
> | GRPO (Baseline)    |        85.7          |
> | Parallel-R1-Seen   |         89.7 (+4.0)  |
> | Parallel-R1-Unseen (S1) |       86.8 (+1.1)            |
> | Parallel-R1-Unseen (S2) |       86.6 (+0.9)            |
>
> > Q1: The study would benefit from a formal analysis of computational efficiency during training.
>
> The Parallel-R1-Seen has the comparable training cost as the sequential baseline: ~3.5 days on 8×40GB GPUs.  The Parallel-R1-Unseen variant requires a longer runtime (\~6 days on the same hardware) because it must dynamically construct multiverse-style 4D attention masks at every RL rollout step. Unlike Multiverse, which preprocesses all 4D masks offline during supervised data construction, our RL setting must build these masks online for every trajectory, resulting in additional overhead. We have added this discussion in Sec. 4.1.
>
> We hope the above clarifications and additional analyses fully address the reviewer’s concerns.

---

### Official Review · Reviewer_iyKL · 2025-11-02

**Soundness:** 2
**Presentation:** 2
**Contribution:** 2
**Rating:** 4
**Confidence:** 5

**Summary:**

Parallel-R1 is an RL framework designed to actually train parallel thinking in LLMs, rather than just prompting it, by letting the model explore multiple reasoning paths on complex tasks. The method uses a progressive curriculum: start with SFT on easy, prompt-generated trajectories to “teach” the parallel pattern, then switch to RL so the model can explore and generalize it on harder problems. The authors evaluate their method over several math benchmarks.

**Strengths:**

1.	The authors propose a framework specifically aimed at improving parallel reasoning in LLMs.
2.	The introduced multiverse attention mechanism enables end-to-end training of parallel reasoning behaviors.
3.	The authors validate the approach with experiments and report performance.

**Weaknesses:**

1.	The authors should clarify why different paths are expected to produce genuinely different reasoning trajectories. As it stands, the only differences between trajectories are the position index and the number of <path> tokens, so it is not obvious that this design actually encourages the model to learn diverse reasoning.
2.	The pass@16 improvement on AIME24 is noticeably smaller than on AIME25; this suggests the method may have high variance across datasets and needs more analysis.
3.	Qwen3-4B is relatively small; it would strengthen the paper to report results on larger models to show the method scales.

**Questions:**

Please check the weakness.

---

> ### Author Response · Authors · 2025-11-27
> **Author Response**
>
> We thank the reviewer iyKL for the constructive and insightful feedback. Below we address all concerns point by point.
>
> > The authors should clarify why different paths are expected to produce genuinely different reasoning trajectories. As it stands, the only differences between trajectories are the position index and the number of [object Object] tokens, so it is not obvious that this design actually encourages the model to learn diverse reasoning.
>
> Thank you for the insightful question. Although all paths share the same prefix, they diverge because each path is decoded through a fully independent autoregressive stream: every path has its own KV cache, no cross-path attention, and uses independent stochastic sampling. Therefore, even a one-token sampling difference at the first parallel step immediately produces different hidden-state, which then amplify into different reasoning paths.
>
> **Empirical validation.** To directly verify whether our parallel thinking paths collapse into the same reasoning chain, we measure both token-level and semantic divergence across paths. We report diversity results using the Parallel-SFT-Unseen and Parallel-R1-Unseen (S2) variants because they exhibit consistently high parallel ratios, ensuring that most examples contain multiple valid paths. As shown below, BLEU scores are extremely low and cosine similarities are far from 1, confirming that paths do not collapse into the same reasoning chain. We have included this analysis in the revised version of our paper. Please find more details about this experiment.
> | Method                 | Token-Level Diversity (BLEU ↓) | Semantic Diversity (Cosine ↓) |  Parallel Ratio |  Acc. |
> |------------------------|--------------------------------|--------------------------------|--------------------------------|--------------------------------|
> | Parallel-SFT-Unseen    |             0.0675                   |            0.6360                    |             95.6                   |            31.7                    |
> | Parallel-R1-Unseen (S2)   |            0.0627                   |           0.6083                    |            63.0                    |            46.8                    |
>
> > The pass@16 improvement on AIME24 is noticeably smaller than on AIME25; this suggests the method may have high variance across datasets and needs more analysis.
>
> We acknowledge that the pass@16 improvement on AIME24 is smaller than on AIME25. Following your suggestion, we further investigate this phenomenon and find that the discrepancy is reasonable, as problems from different subsets naturally have different difficulty and different demand for utilizing parallel thinking.
>
> To provide a more concrete analysis, we conduct a cross-dataset comparison of problem structures. For every pair of problems (A_i from AIME24, B_j from AIME25), we ask an external LLM judge (GPT-4o-mini) to decide which problem is more likely to benefit from parallel thinking. This yields 900 pairwise comparisons in total. To avoid biases, we manually check 50 pairs randomly sampled from these 900 pairs and find the results are consistent with human evaluation. The results are summarized below and indicate AIME25 has a stronger demand for utilizing parallel thinking. This can explain why improvement on AIME24 is noticeably smaller than on AIME25.
>
>
> | Dataset  | Wins | Losses | Total | Win Rate |
> |----------|------|--------|--------|-----------|
> | **AIME25** | **686** | 214 | 900 | **0.762** |
> | **AIME24** | 214 | **686** | 900 | 0.238 |
>
> > Qwen3-4B is relatively small; it would strengthen the paper to report results on larger models to show the method scales.
>
> Thank you for the suggestion. To directly address the concern about scalability, we conducted preliminary experiments on the larger Qwen-2.5-7B-Math backbone.  Under the limited compute and rebuttal-time budget, we train both baseline and Parallel-R1-Unseen (S2) with RL 2 epochs. The results are shown below. These preliminary results provide positive preliminary evidence that our approach scales to larger models.
>
> | Method | AIME24 | AIME25 | AMC23 | MATH | AVG |
> |--------|--------|--------|--------|--------|--------|
> | Qwen-2.5-7B-Math | 24.0 |13.8 |63.4 |79.4 |  45.2   |
> | Parallel-R1-Unseen (S2) |23.3 |16.7 | 65.5 | 83.2 | 47.2|
>
> **We hope the above clarifications and additional analyses fully address the reviewer’s concerns.**

---

### Official Review · Reviewer_EJPr · 2025-11-02

**Soundness:** 3
**Presentation:** 4
**Contribution:** 3
**Rating:** 8
**Confidence:** 3

**Summary:**

A solid RL framework for instilling LLM parallel reasoning.

The paper introducing a new reinforcement learning framework, `Parallel-R1`, to teach LLMs “parallel thinking.” Its motivation is clear and timely, aligning with the surge of reasoning-enhancement research (e.g., R1, Gemini IMO results). The writing is super clear and the organization is easy to follow.

It presents a simple yet effective curriculum RL training with parallel reasoning tags (<Parallel>, <Path>, <Summary>) and a multiverse attention variant.

The work’s significance is strong for mathematical reasoning benchmarks, but broader generalization beyond math remains unclear.

**Strengths:**

1. All components / training method have a clear motivation. I admire authors release such adequate ablation study results and the design motivation behind each component.

2. Strong empirical improvements: Table 2 show 10.2% sequential RL baselines v.s. 42.9% Parallel RL on AIME25

3. Clear training methodology design: Section 3.3’s curriculum RL (SFT → easy RL → hard RL) effectively mitigates cold-start issues.

4. Insightful analysis: Figure 4’s finding that parallel reasoning evolves from exploration to verification is conceptually valuable. We then can use it as a mid-training scaffold sounds promising.

**Weaknesses:**

1. Experiments are restricted to math datasets (AIME, AMC, MATH). I understand that based on the proposed method, the dataset is still hard to obtain. However I still curious about the cross-domain validation.

2, Section 3.4’s “Multiverse” model increases implementation difficulty but the final gain seems very limited. Just want to understand the compute cost.

**Questions:**

1. The curriculum uses an SFT cold-start on easy math (GSM8K) followed by RL on harder tasks. But can you quantify and isolate how much of the final gain comes from the SFT stage vs how much from the RL stage (especially on hard problems)? For example: if you skip the SFT entirely, what happens?

2. Their reward design alternates between a “parallel-structure” reward and an accuracy reward (Section 3.3.2). How sensitive is the final performance to the exact weighting, frequency, and window-size of these alternating rewards? Could a slightly different schedule eliminate the benefit of the “parallel thinking scaffold” effect?

3. The work claims that the model transitions from “parallel thinking for exploration” to “parallel thinking for verification” (Figure 4). How robust is that finding across model sizes, domains (other than math), and random initial seeds? Seems we only conduct experiments on Qwen and math. I wonder if this is applicable to other models and domains.

4. Regarding the “Multiverse” (Parallel-R1-Unseen) architecture: the authors note that performance actually drops in some configurations relative to the autoregressive variant (see Table 2). What hypotheses do authors have about why the explicit path-isolation hurts performance, and how generalizable is that to larger models/harder tasks?

5. The experiments are purely within mathematical reasoning benchmarks (AIME, AMC, MATH). How well would this parallel thinking RL approach transfer to non-numeric reasoning tasks (e.g., commonsense reasoning, planning, multi-hop QA)? What adaptations might be required?

---

> ### Author Response · Authors · 2025-11-27
> **Author Response (1/n)**
>
> **Thank you for taking time to review our paper and for the valuable feedback. We believe these feedback indeed help improve our paper a lot. We are glad to address your questions point by point below.**
>
> > Experiments are restricted to math datasets (AIME, AMC, MATH). I understand that based on the proposed method, the dataset is still hard to obtain. However I still curious about the cross-domain validation.
>
> Thanks for the suggestion. We have added an additional cross domain validation on StrategyQA. The results are shown below. We plan to incorporate a more comprehensive cross-domain validation in the future.
>
> | Method             | StrategyQA Accuracy |
> |--------------------|---------------------------|
> | GPRO (Baseline)    |        85.7          |
> | Parallel-R1-Seen   |         89.7 (+4.0)  |
> | Parallel-R1-Unseen (S1) |       86.8 (+1.1)            |
> | Parallel-R1-Unseen (S2) |       86.6 (+0.9)            |
>
>
> > Section 3.4’s “Multiverse” model increases implementation difficulty but the final gain seems very limited. Just want to understand the compute cost.
>
> Thank you for your question. Yes, “Multiverse” model increases implementation difficulty. The Parallel-R1-Seen has the comparable training cost as the sequential baseline: \~3.5 days on 8×40GB GPUs.  The Parallel-R1-Unseen variant requires a longer runtime (~6 days on the same hardware) because it must dynamically construct multiverse-style 4D attention masks at every RL rollout step. Unlike Multiverse, which preprocesses all 4D masks offline during supervised data construction, our RL setting must build these masks online for every trajectory, resulting in additional overhead.
>
> We agree that, solely from the perspective of final performance gain, the Parallel-R1-Unseen variant exhibits lower effectiveness. However, our primary goal in introducing both variants was not merely to pursue state-of-the-art results, but rather to systematically reveal key insights: By comparing the performance and cost of both architectures, we provide critical architectural insights and highlight challenges for future research on integrating parallel reasoning into LLMs.
>
> > The curriculum uses an SFT cold-start on easy math (GSM8K) followed by RL on harder tasks. But can you quantify and isolate how much of the final gain comes from the SFT stage vs how much from the RL stage (especially on hard problems)? For example: if you skip the SFT entirely, what happens?
>
> Thank you for raising this insightful question. In practice, the SFT and RL stages play complementary but inseparable roles, making it difficult to attribute the final performance to the SFT stage in isolation.
>
> When we skip the SFT cold-start, the model initially struggles to produce meaningful <Parallel> / <Path> structures. As a result, the parallel-structure reward rarely triggers and the training effectively degenerates into sequential RL, without learning interpretable parallel behaviors. Importantly, the SFT stage is not intended to improve accuracy on hard tasks, but only to provide a structural prior so that RL can meaningfully explore in the space of parallel reasoning trajectories. To verify this, we trained this SFT-skipped variant and the results show below.
>
> As our SFT (cold-start) stage just aims to teach basic parallel thinking formats, you can find the SFT model cannot work/generalize well on general math benchmarks. Here RL comes in and enables models to refine and generalize their parallel thinking behaviors to harder problems, therefore improving performance while generating meaningful parallel thinking behaviors. Our Table 2 already shows the role of RL on final performance and we borrow below. For Parallel-R1-Unseen (S2) variant, the RL leads to an performance improvement of 15.1.
>
>
> | Variant           | AIME24 | AIME25 | AMC23 | MATH | Avg | ΔAVG | Parallel Usage |
> |-------------------|--------|--------|--------|--------|----------------|-------------------------------|-------------------|
> | Parallel-R1-Unseen (S2) | 16.3 | 19.0 | 67.5 | 84.5 | 46.8 | - | 63.0% |
> | Parallel-R1-Unseen (S2) - Skip SFT | 18.8 | 15.0 | 62.8 | 83.9 |45.1 |-1.7 |~0.0% |
> | Parallel-SFT-Unseen | 8.5     | 5.2 | 41.7 | 71.5 | 31.7  | - 15.1 | 95.6% |

---

> > ### Author Response · Authors · 2025-11-27
> > **Author Response (2/n)**
> >
> > > Their reward design alternates between a “parallel-structure” reward and an accuracy reward (Section 3.3.2). How sensitive is the final performance to the exact weighting, frequency, and window-size of these alternating rewards? Could a slightly different schedule eliminate the benefit of the “parallel thinking scaffold” effect?
> >
> > Thank you for the thoughtful question. We agree that understanding the sensitivity of the alternating reward schedule is important. We provide additional diagnostic experiments and analysis below.
> >
> > (1) Sensitivity Analysis: Performance is robust.
> >
> > We tested several reasonable variations of the schedule on Parallel-R1-Unseen (S2), including changes to
> >  • parallel-vs-accuracy frequency (e.g., 8/2, 6/4),
> >  • weighting (1.2, 2.0), and
> >  • window-size (10–20).
> > Our Parallel-r1-Unsee (S2) is robust across different weight, window size and frequency.
> > ### Reward Sensitivity (weight, window size and frequency)
> > | Setting                                | AIME24 | AIME25 | AMC23 | MATH |
> > |----------------------------------------|--------|--------|--------|-------|
> > | weight = 1.2 (Parallel-R1-Unseen (S2))              |      16.3    |     19.0    |    67.5    |    84.5   |
> > | weight = 2.0 (Stronger struct. bias) |   17.7     |    17.9    |    64.5    |  84.2     |
> > | Window-size = 10 (Parallel-R1-Unseen (S2))        |    16.3    |     19.0    |    67.5    |    84.5   |
> > | Window-size = 20              |   17.5     |   19.4    |    65.3    |   84.2    |
> > | Frequency = 8/2 (Parallel-R1-Unseen (S2)) |    16.3    |     19.0    |    67.5    |    84.5   |
> > | Frequency = 6/4        |   18.2     |    19.6    |   65.6     |   83.9    |
> >
> > (2) Why the scaffold does not disappear under different schedules
> >
> > We can find under all reward settings, models can learn stable parallel thinking behaviors, e.g., parallel ratio > 0.6,  therefore, we expect a slightly different schedule cannot eliminate the benefit of the “parallel thinking scaffold” effect.
> >
> > > The work claims that the model transitions from “parallel thinking for exploration” to “parallel thinking for verification” (Figure 4). How robust is that finding across model sizes, domains (other than math), and random initial seeds? Seems we only conduct experiments on Qwen and math. I wonder if this is applicable to other models and domains.
> >
> > Thank you for raising this question. We measure the trigger position of parallel thinking behaviors of Parallel-R1-Unseen(S2) (Qwen2.5-7B-Math) and Parallel-R1-Unseen(S2) on StrategyQA and can consistently observe the phenomenon——model transitions from “parallel thinking for exploration” to “parallel thinking for verification”.
> >
> > ### Mean Relative Position (Selected Steps)
> >
> > | Model   | 20      | 40      | 60 | 80 | 100 | 120 |
> > |---------|---------|---------|----|----|-----|-----|
> > | Parallel-R1-Unseen(S2) (Qwen2.5-7B-Math)  | 0.4652  | 0.5269  | 0.5707  | 0.5964  | 0.6364   | 0.6508   |
> >
> > | Dataset     | 20     | 200    |
> > |-------------|--------|--------|
> > | StrategyQA  | 0.3875 | 0.5464|

---

> > > ### Author Response · Authors · 2025-11-27
> > > **Author Response (3/n)**
> > >
> > > > Regarding the “Multiverse” (Parallel-R1-Unseen) architecture: the authors note that performance actually drops in some configurations relative to the autoregressive variant (see Table 2). What hypotheses do authors have about why the explicit path-isolation hurts performance, and how generalizable is that to larger models/harder tasks?
> > >
> > > Thanks for raising this insightful question. Our main hypothesis is that the performance degradation comes from a misalignment between pre-training behavior and post-training modifications.
> > >
> > > LLMs are pre-trained on large-scale corpora purely in an autoregressive manner, where all previously generated content remains visible. The Parallel-R1-Seen variant preserves this autoregressive training objective, making it more aligned with the model’s pre-training objective, which explains its stronger performance. In contrast, Parallel-R1-Unseen introduces explicit path isolation by modifying the attention mechanisms so that different reasoning paths cannot see each other. This architectural/objective shift deviates from how the model was originally pre-trained, and post-training alone may not be sufficient to fully overwrite these long-established autoregressive behaviors.
> > >
> > > Larger models, which have better learning ability, could be more effective to alleviate such misalignment, therefore alleviate the performance gap. Additionally, we observe similar phenomenon in StrategyQA task.
> > >
> > > > The experiments are purely within mathematical reasoning benchmarks (AIME, AMC, MATH). How well would this parallel thinking RL approach transfer to non-numeric reasoning tasks (e.g., commonsense reasoning, planning, multi-hop QA)? What adaptations might be required?
> > >
> > > Thanks for raising this insightful question.  Currently, we focus on mathematical reasoning because it is a classic kind of real-world complex reasoning task and provides a clean and deterministic correctness signal, which is crucial for RL. This is also consistent with prior RL-for-reasoning work.
> > >
> > > Conceptually, we view the proposed parallel thinking RL framework as task-general rather than math-specific. Our definition of parallel thinking process, which consists of parallel exploration and summary, is general for all kinds of tasks and can naturally apply to commonsense reasoning, planning, and multi-hop QA.
> > >
> > > **Test-time adaptation**: we can directly evaluate on other domains using the existing checkpoint. The results on StrategyQA are shown above.
> > >
> > > **Training-time adaptation**: Extending to non-numeric tasks mainly requires adapting the reward signal and the potentially SFT data construction. The underlying RL mechanism, parallel structure, and training framework remain unchanged.
> > >
> > > **We hope the above clarifications and additional analyses fully address the reviewer’s concerns.**

---

### Official Review · Reviewer_CufE · 2025-11-03

**Soundness:** 1
**Presentation:** 2
**Contribution:** 2
**Rating:** 2
**Confidence:** 5

**Summary:**

The paper proposes Parallel-R1, a reinforcement learning framework that allows reasoning in parallel for complex real-world reasoning tasks. Specifically, it introduces a two-stage framework that performs SFT on prompt-generated trajectories from easier tasks to instill parallel thinking and then RL to explore and generalize on harder problems. Experiments show that the proposed method achieves improved performance on sequential RL baselines.

**Strengths:**

* The paper is the first RL framework to learn parallel thinking on general reasoning tasks. Prior works either only use SFT or perform RL on tasks with synthetic data.
* The work offers a complete recipe, including SFT and RL, to elicit parallel thinking. The recipe is compatible with existing popular reinforcement learning pipelines such as DAPO.
* Benchmarks show performance improvements over sequential RL baselines.

**Weaknesses:**

Comparisons with prior works:

* No comparison with prior work Multiverse [1] that the work is heavily based on. Specifically, this work reuses the handing of attention masks and position ids from Multiverse (Sec 3.4). A comparison is needed since both the baseline and the proposed method use distilled reasoning trajectories from a reasoning model such as DeepSeek-R1.
* The performance is much worse than prior works that do not use RL. Multiverse achieves 53.8% on AIME 24 (Table 2 in [1]), while the work achieves 19.0% (with Parallel-R1-Unseen). This work does not motivate why RL is necessary for parallel thinking, since prior works with only SFT is able to achieve much higher performance. It is true that Multiverse is based on Qwen2.5-32B-Instruct and this work is based on Qwen3-4B-Base, but the cost of running SFT on Qwen2.5-32B-Instruct is much lower than running SFT and RL on Qwen3-4B-Base.
* The comparison is also much worse than Qwen3-4B, the sequential reasoning variant of the base model. Qwen3-4B achieves 76.0% on AIME 24 (Table 17 in [2]), which is much higher than the work's result. This makes the method less useful in real-world applications, as the performance demonstrated with the proposed parallel reasoning recipe is sub-par compared to existing sequential reasoning variant.
* The evaluation for Qwen3-4B-Base is likely problematic. Qwen3-4B-Base achieves 54.10% in MATH without instruction tuning (Table 7 in [2]), which is much higher than the work's result, which is 13.9%.

Data pipeline:

* In Sec 3.2, the authors mentioned that the LLM pipeline in Multiverse is computationally intensive and limited scalability. However, the proposed data pipeline also involves an LLM. Although the LLM used in the data pipeline is smaller than the one in Multiverse, the pipeline does not work with complex problems (e.g., those in DAPO, as indicated in Table 1). The advantage of the proposed data pipeline is not justified, as Multiverse is able to handle long trajectories with complex problems in the s1K-1.1 dataset.

Performance:

* The method's baseline selection is not justified. An improved sequential reasoning baseline is to train Qwen3-4B-Base model with the proposed SFT dataset on GSM8K as a sequential model, treating the added special tokens as typical text tokens, and then perform DAPO on the model in a fair setting. The evaluation should be performed using a sequential inference engine. The reviewer believes this is different from "Parallel-R1-Seen", since "Parallel-R1-Seen" is inferenced with a parallel inference engine.
* In Table 1, the "Parallel-R1-Seen", an autoregressive method (L342) that does not prevent paths from seeing each other in training, outperforms parallel methods (S1 and S2) by a large margin. Furthermore, there might be a discrepancy between training and inference, as the paths are generated in parallel at inference.

Efficiency:

* The method is proposed to improve both performance and efficiency (L129, L300). However, no efficiency comparisons are provided with the baseline methods. The # Parallel is not an efficiency metric because the model can output much longer sequences in parallel to achieve the same performance as in sequential, causing higher latency than in sequential. A valid comparison metric is the latency in terms of the tokens, or wall-clock time if an efficient inference engine is used.

Minor point:

* Writing is not clear: abstract and intro mentions a 42.9% improvement over sequential RL baselines, but this number is not reported in the experiments.
* In Fig. 4, the parallel ratio goes to 0 starting at Step 300, but the parallel ratio is non-zero in the Parallel-R1 results in Table 2. In L308, the authors indicated that the model is trained with 300 gradient update steps in Stage 2 (which is around Step 500 in total in Fig. 4, as there is a 200-step RL training in Stage 1). These two results are inconsistent. Furthermore, if training for 300 steps leads to a parallel ratio of 0, it indicates that the model will not be able to perform parallel reasoning at inference and will not have improved efficiency compared to the sequential baseline.

[1] Multiverse: Your Language Models Secretly Decide How to Parallelize and Merge Generation. Yang, et al. https://arxiv.org/abs/2506.09991

[2] Qwen3 Technical Report. Qwen Team. https://arxiv.org/abs/2505.09388

**Questions:**

In addition to the questions in the weaknesses section, the reviewer has the following questions:

1. What are the training time and hardware requirements for Parallel-R1-Unseen (4B)? How does it compare to Multiverse-32B?
2. What is the exact inference setting for "Parallel-R1-Seen"? Is it with a parallel inference engine?
3. Are there any interpretations for going fully sequential after 300 steps in Fig. 4?

---

> ### Author Response · Authors · 2025-11-26
> **Author response (1/n)**
>
> > No comparison with prior work Multiverse [1] that the work is heavily based on. Specifically, this work reuses the handing of attention masks and position ids from Multiverse (Sec 3.4). A comparison is needed since both the baseline and the proposed method use distilled reasoning trajectories from a reasoning model such as DeepSeek-R1.
>
> **Although Parallel-R1 draws inspiration from Multiverse, its novelty lies fundamentally different.
> Parallel-R1 introduces the first RL framework designed to cultivate parallel thinking for complex, real-world reasoning tasks. Achieving this is far from trivial – it requires overcoming two core challenges: (1) developing effective cold-start strategies, and (2) designing robust reward models.**
>
> We provide a detailed comparison between our work and Multiverse below:
>
> **Multiverse** demonstrated that parallel reasoning for real-world tasks is possible through SFT by mimicking parallel thinking demonstrations from strong teachers (Deepseek and Gemini-2.5 Pro). However, this reliance on high-quality teacher traces restricts its scalability and limits the model to the teacher's upper bound. Sometimes, the model cannot learn meaningfully parallel thinking behaviors and has poor generalization.
>
> **Parallel-R1** is a scalable RL-driven exploration paradigm. Our core contribution is shifting the learning process from imitation to autonomous exploration, allowing the model to learn parallel thinking behaviors via exploration starting from cold-start. Notably, our cold-start dataset only aims to teach format instead of reasoning. Our work is the first to systematically investigate this in real-world tasks, introducing (1) effective cold-start strategies, and (2) robust reward modeling. Notably, with the goal of providing more insights to the research community, we systematically investigate two distinct structural variants: the autoregressive model and the Multiverse model.
>
> The different goal of using “distilled” reasoning trajectories: We also wish to correct the premise that "both the baseline and the proposed method use distilled reasoning trajectories."
>
> Multiverse relies on Knowledge Distillation to mimic the reasoning logic of a strong teacher (Deepseek) while learning parallel thinking behaviors.
>
> Parallel-R1 uses cold-start data only to initialize the structural format. The reasoning logic is acquired entirely through RL exploration. Therefore, the premise that both methods rely on the same supervision signal is incorrect.
>
> **Empirical comparison with Multiverse: RL training requires significantly more computational resources than SFT. To ensure a fair comparison with Multiverse, we reproduce its results using Qwen3-4B-Base.**
>
> **We also emphasize that Parallel-R1’s contribution is orthogonal to that of Multiverse. The RL training can be built on top of Multiverse’s SFT pipeline, forming a progression from Base model → SFT model (Multiverse) → RL-trained model (Parallel-R1).**
>
> | Method              | AIME24 (Mean@16) | AIME25 (Mean@16) | AMC23 (Mean@16) | MATH (Mean@1) |
> |---------------------|------------------|------------------|------------------|----------------|
> | Multiverse-SFT-4B      |       8.2      |       10.6       |      49.0       |      76.8      |
> | Sequential RL Baseline |  18.5              | 14.8              |  63.6              | 83.5            |
> | Parallel-R1-Seen    | 19.4              | 19.2              | 70.5              | 86.7            |
> | Parallel-R1-Unseen (S1)  |  18.3              | 17.7              |  69.7              | 82.6            |
> | Parallel-R1-Unseen (S2)  | 16.3              | 19.0              |  67.5              | 84.5            |

---

> > ### Comment · Reviewer_CufE · 2025-11-26
> >
> > ## Difference between Parallel-R1 and Multiverse
> >
> > Thanks for offering this clarification. The reviewer does understand the difference between Parallel-R1 and Multiverse is the additional RL stage, but the claim in the response is still not convincing. The reviewer explains the rationale behind this conclusion below:
> >
> > > Our core contribution is shifting the learning process from imitation to autonomous exploration, allowing the model to learn parallel thinking behaviors via exploration starting from cold-start. Notably, our cold-start dataset only aims to teach format instead of reasoning.
> >
> > The response to the reviewer's concern indicates that SFT cold-start in Parallel-R1 is only for learning format instead of reasoning. However, the reviewer's concern is still not fully addressed, because it does not explain why the SFT cold-start for format only is better than the SFT cold-start for reasoning in addition to format, which was employed in Multiverse. It seems that the fact that the SFT cold-start for format only is a limitation of the data curation pipeline in Parallel-R1 instead of a feature of the Parallel-R1 framework.
> >
> > In addition, the data curation pipeline in Multiverse allows learning both, while Parallel-R1 only learns format. Furthermore, the data curation pipeline in Multiverse allows much more complex reasoning tasks (e.g., s1 dataset), while Parallel-R1 only learns format with simple reasoning tasks in GSM8k, and is demonstrated to not be able to work on complex problems (e.g., DAPO), as demonstrated in Table 1 of the Parallel-R1 paper.
> >
> > The reviewer still has the concern that the pipeline in Parallel-R1 is inferior to the pipeline in Multiverse due to the problems it could process, and the ability to produce data that allows learning both format and reasoning.
> >
> > > Multiverse relies on Knowledge Distillation to mimic the reasoning logic of a strong teacher (Deepseek) while learning parallel thinking behaviors.
> > > Parallel-R1 uses cold-start data only to initialize the structural format. The reasoning logic is acquired entirely through RL exploration. Therefore, the premise that both methods rely on the same supervision signal is incorrect.
> >
> > It's true that Multiverse learns reasoning knowledge from Deepseek, but Parallel-R1 does not demonstrate the benefit for not learning reasoning knowledge from Deepseek. There are abundant trajectories from Deepseek or other open reasoning models available to train, so it's not a technical difficulty. In addition, the Deepseek paper shows that cold-start with long CoT does help a lot compared to just learning reasoning with RL alone (https://arxiv.org/abs/2501.12948, Fig. 1 and Fig. 2).
> >
> > Finally, the rebuttal response claims that "The reasoning logic is acquired entirely through RL exploration." This itself is not justified, because the data is also generated from Deepseek-Qwen3-8B in Parallel-R1, so it still could introduce reasoning knowledge into the model.
> >
> > > We also emphasize that Parallel-R1’s contribution is orthogonal to that of Multiverse.
> >
> > In summary, the data pipeline also uses an LLM (deepseek-qwen3-8b) in data curation, so the technical requirements are similar for data pipeline compared to Multiverse, but Multiverse could process more complex reasoning tasks with reasoning knowledge in the curated trajectories, while Parallel-R1 only learns format with simple reasoning tasks. This contradicts with the claims that the contributions are orthogonal: you cannot apply both techniques for data curation in one pipeline, since the "data for format learning" is one of the two key contributions of Parallel-R1.

---

> > > ### Author Response · Authors · 2025-12-02
> > > **Further Clarification on Difference between Parallel-R1 and Multiverse (1/n)**
> > >
> > > Thank you for raising additional concerns regarding the distinction between our Parallel-R1 framework and the Multiverse approach. We recognize that the reviewer’s comments center around one core concern:  why Parallel-R1 adopts a format-only SFT cold start instead of a Multiverse-style format+reasoning SFT, and whether this design reflects a pipeline limitation. All other detailed comments appear to stem from this underlying issue. We address this core concern below and then briefly comment on several remaining points for completeness.
> > >
> > > ## **1. Why we do not use Multiverse-style SFT as cold-start**
> > >
> > > Multiverse-style SFT relies on long CoT trajectories from strong teacher models and a multi-stage rewriting pipeline that converts sequential CoTs into a parallel structure. **While highly effective for imitation learning, these data pipelines are designed to transfer teacher-provided reasoning behaviors** and therefore serve a different purpose from our goal of studying **RL-driven emergence**.
> > >
> > > Critically, rewritten long CoTs are *not* native parallel reasoning. They are sequential trajectories reformatted into a parallel structure. Using such pseudo-parallel demonstrations as cold-start supervision would **inject teacher-induced non-native parallel reasoning strategies before RL**, pre-shaping the model’s behaviors and fundamentally misaligning with our goal of examining how parallel thinking can *emerge* through RL exploration in a pure parallel environment.
> > >
> > > > Importantly, this is not a pipeline limitation but a methodological requirement.
> > > > Using Multiverse-style SFT would inject non-native parallel-thinking behaviors that are *not* learned under a pure parallel-reasoning setting, which would obscure the phenomenon we aim to study and undermine the purpose of examining RL-driven emergence.
> > >
> > > To avoid this, our cold start is deliberately designed to primarily provide structural priors for the parallel-thinking format, with minimal and shallow reasoning content (short CoT) drawn from easy tasks (e.g., GSM8K). This setup provides the structural priors necessary for RL to effectively explore parallel-thinking behaviors, without pre-imposing complex teacher reasoning patterns.
> > >
> > > **We have revised Sec. 3.2 to clarify this part and ensure the intended methodological motivation is more explicit.**
> > > ## **2. On the concern that our SFT pipeline handles only simple tasks**
> > >
> > > This concern applies an evaluation metric that is not aligned with our research objective. The purpose of the SFT cold start in Parallel-R1 is **not** to solve complex reasoning tasks; the purpose is to enable **RL to explore parallel-thinking behaviors effectively**.
> > >
> > > Under this correct metric, the pipeline is effective: Table 2 shows that Parallel-R1 variants trained from this initialization **outperform several sequential RL baselines on complex mathematical reasoning benchmarks**, demonstrating that the cold-start design fulfills its role.
> > >
> > > Thus, the simplicity of the SFT data is an intentional methodological choice, not a limitation.
> > >
> > > ## **3. On the point that both pipelines use LLMs and therefore have similar technical requirements**
> > >
> > > Using an LLM is not the technical bottleneck.
> > >  **What differs is the data objective:**
> > >
> > > - Multiverse construct *long parallel trajectories*, requiring multi-step semantic transformations such as tree parsing, node detection, MapReduce-style restructuring, and trajectory refilling.
> > > - Parallel-R1 requires generating *short, format-oriented outputs* on easy tasks without any structural rewriting.
> > >
> > > The pipelines therefore differ fundamentally in complexity and purpose.
> > >
> > > ## **4. On Clarifying the RL–SFT Attribution Wording**
> > >
> > > We appreciate the reviewer’s clarification and have refined our wording.
> > > To avoid any potential misunderstanding, we note that the phrasing “entirely through RL” only appeared in our initial rebuttal wording and **is not part of the paper’s claims**. The main paper consistently avoids making such an absolute statement.
> > >
> > > A more accurate characterization of our setting is:
> > >
> > > > **“The reasoning logic is acquired primarily through RL exploration.”**
> > >
> > > This wording is fully aligned with the methodological constraints described in the paper: the SFT data intentionally contains only easy tasks (e.g., GSM8K), shallow reasoning (short CoT), and structural format supervision, which limits the presence of complex teacher reasoning patterns prior to RL.
> > >
> > > Thus, this refinement does not change the paper’s technical content but clarifies the intent of our earlier rebuttal phrasing.

---

> ### Author Response · Authors · 2025-11-26
> **Author Response (2/n)**
>
> > The performance is much worse than prior works that do not use RL. Multiverse achieves 53.8% on AIME 24 (Table 2 in [1]), while the work achieves 19.0% (with Parallel-R1-Unseen). This work does not motivate why RL is necessary for parallel thinking, since prior work with only SFT is able to achieve much higher performance. It is true that Multiverse is based on Qwen2.5-32B-Instruct and this work is based on Qwen3-4B-Base, but the cost of running SFT on Qwen2.5-32B-Instruct is much lower than running SFT and RL on Qwen3-4B-Base.
>
>
> **The primary reason for the performance gap is that Multiverse is based on a 32B model, whereas our experiments use a 4B model. Because RL training requires significantly more GPU resources than SFT, we are unable to train such a large model with RL in our lab. To address this and enable a fair comparison, we reproduced Multiverse on a 4B model, as shown in the table above. From these results, we can clearly see that adding an additional RL stage improves performance.**
>
> This observation is consistent with many recent LLM post-training practices, such as from Qwen and Deepseek: applying RL after SFT generally yields further gains, since RL training enables greater exploration and on-policy optimization.
>
> Regarding the cost, we acknowledge that RL training time is indeed longer than SFT. However, evaluating cost solely based on training FLOPs is not the whole story. In SFT, Data Curation is also costly. For example, Multiverse relies on a complex, expensive pipeline to synthesize traces using proprietary frontier models (e.g., Gemini-2.5-Pro). Scaling up the SFT paradigm requires linearly scaling this expensive data generation process, leading to exploding costs and dependency on closed-source APIs. While RL requires more training FLOPs, it eliminates the need for expensive teacher-generated data. RL offers a scalable path where the model improves through self-exploration, whereas SFT is strictly bottlenecked by the cost and capability of the teacher.
>
>
> > The comparison is also much worse than Qwen3-4B, the sequential reasoning variant of the base model. Qwen3-4B achieves 76.0% on AIME 24 (Table 17 in [2]), which is much higher than the work's result. This makes the method less useful in real-world applications, as the performance demonstrated with the proposed parallel reasoning recipe is sub-par compared to existing sequential reasoning variant.
>
> We clarify that it is not rigorous to derive that parallel reasoning is less useful in real-world applications by comparing our Parallel-R1 (19.0%) directly with the product-level Qwen3-4B (76.0%). Qwen3-4B’s high performance comes from strong-to-weak distillation from flagship LLMs and massive post-training data. By contrast, our Parallel-R1 starting from Qwen-3-4B base with nearly 7K cold-start data on easy math and 17K RL data from DAPO. The performance gap can only show the success of scaling up sequential reasoning instead of parallel reasoning is less useful.
>
> A more rigorous setting to show the effectiveness/usefulness of parallel reasoning to compare with sequential baselines with same data and same algorithms. In our paper, we have added two strong sequential baselines: 1) GRPO on DAPO and 2) GRPO on GSM8K and then DAPO. We can find our Parallel-R1-variants can consistently outperform them, indicating that parallel thinking is useful.
>
> These results highlight the potential of parallel reasoning as a new scaling dimension for pushing the performance boundary of reasoning models. Our paper, which provides the first RL framework to learn parallel reasoning for general tasks, provides a scalable path towards this promising goal.

---

> > ### Comment · Reviewer_CufE · 2025-11-26
> >
> > ## Performance Gap between Parallel-R1 and Multiverse
> >
> > > The primary reason for the performance gap is that Multiverse is based on a 32B model, whereas our experiments use a 4B model.
> >
> > Thanks for offering this clarification that the gap is due to different model capacities. However, the reviewer believes the gap is not due to the model capacity difference, but due to the difference in the data curation pipeline.
> >
> > Specifically, Qwen3-4B, the reasoning model offered by Qwen, performs much better compared to Parallel-R1 (Qwen3-4B). Specifically, while Qwen3-4B (2507) obtains 47.4% on AIME25 (https://huggingface.co/Qwen/Qwen3-4B-Instruct-2507), Parallel-R1 obtains 19.2% only. Multiverse-32B, which is much larger, obtains 45.8 (https://arxiv.org/abs/2506.09991), which is still lower than Qwen3-4B.
> >
> > To the best of the reviewer's knowledge, Qwen3-4B (the reasoning model post-trained on Qwen3-4B-Base) is trained with SFT only, using knowledge from larger Qwen3 models (https://arxiv.org/abs/2505.09388). Please correct the reviewer if this is incorrect. This disproves the effectiveness of having to leverage RL for effective reasoning. Since Parallel-R1 claims to use RL for effective reasoning (while SFT is for format only), this further weakens the claim that Parallel-R1 is more effective than prior work based on SFT only (e.g., Multiverse).
> >
> > > This observation is consistent with many recent LLM post-training practices, such as from Qwen and Deepseek: applying RL after SFT generally yields further gains, since RL training enables greater exploration and on-policy optimization.
> >
> > It's true that applying RL after SFT generally yields further gains, and there are works that apply RL upon SFT models (e.g., DeepScaleR/RLLM, https://github.com/rllm-org/rllm). However, the authors did not demonstrate effectiveness on any such training setup to justify the claim.
> >
> > > The primary reason for the performance gap is that Multiverse is based on a 32B model, whereas our experiments use a 4B model. Because RL training requires significantly more GPU resources than SFT, we are unable to train such a large model with RL in our lab.
> >
> > There are known works that train smaller models with RL after SFT. Specifically, *POLARIS: A POst-training recipe for scaling reinforcement Learning on Advanced ReasonIng modelS*, a pipeline with 600+ stars on GitHub, trains a model (Qwen3-4B) with exactly the same size as the one used in Parallel-R1, and obtains 79.4\% accuracy on AIME25, which is much higher than Parallel-R1 (19.2\%). This disproves the claim that it is because of the model size difference that Parallel-R1 is not able to start with a model trained after standard post-training.
> >
> > > We clarify that it is not rigorous to derive that parallel reasoning is less useful in real-world applications by comparing our Parallel-R1 (19.0%) directly with the product-level Qwen3-4B (76.0%). Qwen3-4B’s high performance comes from strong-to-weak distillation from flagship LLMs and massive post-training data.
> > > A more rigorous setting to show the effectiveness/usefulness of parallel reasoning to compare with sequential baselines with same data and same algorithms.
> >
> > The reviewer agrees that because the work starts from Qwen3-4B-Base, it can be hard to train with massive data to compete with production-level models. If this is the case, then directly comparing with Polaris on Qwen3-4B that inherits the reasoning ability from massive post-training data will be very valuable. This is because one hypothesis is that running SFT to learn parallel format (as proposed to be an important contribution of Parallel-R1) will lead to catastrophic forgetting of the reasoning ability. If this is true, the proposed contribution will be undermined, and an experiment with SFT on the proposed GSM8k-parallel and GRPO with Polaris data will be able to demonstrate if this is the case.
> >
> > > Multiverse-SFT-4B achieves 10.6\% on AIME25.
> >
> > The reviewer would like to thank the authors for providing this comparison, but Multiverse recipe is not tailored to a base model (and it's dataset is much smaller because they start from Qwen-2.5-32B-Instruct). They demonstrate that they are compatible with a model that has already been trained with massive data, which has not been demonstrated in Parallel-R1. Since Polaris could improve on Qwen3-4B, it would be valuable to directly compare with Polaris on Qwen3-4B to demonstrate the effectiveness of Parallel-R1's format-only cold-start and stable reward designs.

---

> > > ### Author Response · Authors · 2025-12-03
> > > **Further Clarification on Performance Gap between Parallel-R1 and Multiverse**
> > >
> > > > Parallel-R1 underperforms Qwen3-4B reasoning; maybe your pipeline is weak/RL is unnecessary.
> > >
> > > Qwen3-4B-reasoning comes from Qwen3-4B-Base + strong-to-weak distillation ; Parallel-R1 comes from Qwen3-4B-Base + cold-start  + RL . These pipelines differ fundamentally in their data source, data scale, and training design. Therefore, the two systems are not in a comparable experimental setting, and performance differences between them cannot be used to infer whether RL is necessary.
> > >
> > > > It's true that applying RL after SFT generally yields further gains, and there are works that apply RL upon SFT models (e.g., DeepScaleR/RLLM, https://github.com/rllm-org/rllm). However, the authors did not demonstrate effectiveness on any such training setup to justify the claim.
> > >
> > > Our statement that “RL after SFT generally yields further gains” refers to a widely observed trend in recent LLM post-training pipelines (e.g., DeepSeek, Qwen, DeepScaleR/RLLM), and is intended as contextual motivation rather than an empirical claim demonstrated by our paper.
> > >
> > > > POLARIS: A POst-training recipe for scaling reinforcement Learning on Advanced ReasonIng modelS, a pipeline with 600+ stars on GitHub, trains a model (Qwen3-4B) with exactly the same size as the one used in Parallel-R1, and obtains 79.4% accuracy on AIME25, This disproves the claim that it is because of the model size difference that Parallel-R1 is not able to start with a model trained after standard post-training.
> > >
> > > POLARIS and Qwen3-4B both focus on sequential reasoning. Given the same objective, POLARIS can start from Qwen3-4B to further optimize sequential reasoning and push the performance. However our Parallel-R1 studies parallel reasoning, which is different from sequential reasoning. Learning parallel reasoning behaviors starting from a model already optimized for sequential reasoning is not reasonable, as they are different reasoning behaviors. Importantly, Qwen3-4B-Reasoning itself is trained from the base model. Therefore, studying a new reasoning strategy, e.g., parallel reasoning, should also start from a base model, instead of a model with mismatched objectives.
> > >
> > > >The reviewer agrees that because the work starts from Qwen3-4B-Base, it can be hard to train with massive data to compete with production-level models. If this is the case, then directly comparing with Polaris on Qwen3-4B that inherits the reasoning ability from massive post-training data will be very valuable. This is because one hypothesis is that running SFT to learn parallel format (as proposed to be an important contribution of Parallel-R1) will lead to catastrophic forgetting of the reasoning ability. If this is true, the proposed contribution will be undermined, and an experiment with SFT on the proposed GSM8k-parallel and GRPO with Polaris data will be able to demonstrate if this is the case.
> > >
> > > Our Parallel-R1 studies parallel reasoning, which is different from sequential reasoning. We expect not to train parallel reasoning upon a sequential reasoning model. Additionally, given the fact that early sequential reasoning, e.g., Qwen3-4B/Deepseek-R1-Zero are all trained from a base model, a parallel reasoning model should also be trained starting from a base model, instead of a sequential reasoning model.
> > >
> > > As a parallel reasoning model should be trained starting from a base model, applying our SFT will not lead to catastrophic forgetting of the reasoning ability as the base model does not have strong reasoning ability.
> > >
> > > > The reviewer would like to thank the authors for providing this comparison, but Multiverse recipe is not tailored to a base model (and it's dataset is much smaller because they start from Qwen-2.5-32B-Instruct). They demonstrate that they are compatible with a model that has already been trained with massive data, which has not been demonstrated in Parallel-R1. Since Polaris could improve on Qwen3-4B, it would be valuable to directly compare with Polaris on Qwen3-4B to demonstrate the effectiveness of Parallel-R1's format-only cold-start and stable reward designs.
> > >
> > > We respectfully clarify that Parallel-R1 does not view parallel reasoning as a component to be “embedded” into a strong sequential reasoning model. Instead, our work treats parallel reasoning as an independent reasoning paradigm, on par with sequential reasoning, and studies using RL to induce this capability.
> > >
> > > While evaluating compatibility with strong sequential models is an interesting direction, it constitutes a different research problem from the one addressed in this paper. We therefore include it in the Limitation and Future Work section as a potential extension, but it lies outside the scientific focus of Parallel-R1.

---

> ### Author Response · Authors · 2025-11-26
> **Author Response (3/n)**
>
> > The evaluation for Qwen3-4B-Base is likely problematic. Qwen3-4B-Base achieves 54.10% in MATH without instruction tuning (Table 7 in [2]), which is much higher than the work's result, which is 13.9%.
>
> The discrepancy arises because base models often fail to produce answers in the strict output format required by the rule-based verifier. In contrast, the Qwen3 technical report evaluates the base model using an LLM-judge, which is more tolerant of formatting issues. Following the evaluation code used in the General-Reasoner [1] paper, we re-evaluated Qwen3-4B-Base accordingly.
>
> AIME’25: 1.3 → 5.5 (mean@16), 10.2 -> 24.6 (pass@16)
>
>
> AIME’24: 2.9 → 10.0 (mean@16), 16.5 -> 25.4 (pass@16)
>
>
> AMC: 8.1 → 39.3 (mean@16), 51.2 -> 74.6 (pass@16)
>
>
> MATH: 13.9 → 54.0
>
>
> [1] General-Reasoner: Advancing LLM Reasoning Across All Domains. NeurIPS 2025
>
>
> **We updated our PDF accordingly.**
>
>
>
> > In Sec 3.2, the authors mentioned that the LLM pipeline in Multiverse is computationally intensive and limited scalability. However, the proposed data pipeline also involves an LLM. Although the LLM used in the data pipeline is smaller than the one in Multiverse, the pipeline does not work with complex problems (e.g., those in DAPO, as indicated in Table 1). The advantage of the proposed data pipeline is not justified, as Multiverse is able to handle long trajectories with complex problems in the s1K-1.1 dataset.
>
>
> **We acknowledge “computationally intensive” is not an appropriate statement, we removed this phrase in the updated PDF.**
>
> For the scalability statement. Multiverse requires high-quality SFT data, fully solved trajectories for every problem. This leads to fundamentally limited scalability: each additional training example requires expensive teacher generation. In contrast, our pipeline requires only simple cold-start data to learn format and places the reasoning burden on RL rather than on teacher data. Because structural-format data is cheap to produce, the pipeline can scale to arbitrarily large datasets at reasonable cost. This is a qualitatively different scalability profile from Multiverse.
>
> We also respectfully clarify that our cold-start dataset is not designed to solve complex problems such as those in DAPO. Instead, its sole purpose is to teach the structural format of parallel thinking (e.g., how to produce <Parallel> and <Path> blocks), so that RL can explore and learn the reasoning behaviors itself. Evaluating the cold-start data on DAPO is therefore misaligned with its intended function.
>
> > The method's baseline selection is not justified. An improved sequential reasoning baseline is to train Qwen3-4B-Base model with the proposed SFT dataset on GSM8K as a sequential model, treating the added special tokens as typical text tokens, and then perform DAPO on the model in a fair setting. The evaluation should be performed using a sequential inference engine. The reviewer believes this is different from "Parallel-R1-Seen", since "Parallel-R1-Seen" is inferenced with a parallel inference engine.
>
> Thank you for your suggestions. To ensure a fully fair comparison, we trained the improved sequential reasoning baseline exactly following the reviewer’s protocol: Qwen3-4B-Base (without adding special tokens) → parallel-thinking SFT (treating special tokens as plain text) → GRPO on DAPO ( sequential inference engine).The results are presented below.
>
> This baseline is indeed stronger than the original sequential RL baselines, confirming the reviewer’s intuition. However, it still consistently underperforms all Parallel-R1 variants across AIME24, AIME25, AMC23, and MATH.
>
> This emphasizes that the effectiveness of Parallel-R1 arises from a coherent full-loop parallel reasoning design. This design integrates: (i) cold-start parallel-thinking data, (ii) structured parallel-thinking prompts, and (iii) a parallel inference mechanism that enables multi-path exploration in parallel. These components are tightly aligned and collectively define the parallel reasoning paradigm. We believe our full loop design can serve a strong foundation for further work on RL for Parallel Reasoning.
>
> | Model                  | AIME25 Mean@16 | AIME25 Mean@16 | AIME24 Mean@16 | AIME24 Mean@16 | AMC23 Mean@16 | AMC23 Mean@16 | MATH Mean@1 | Avg. |
> |------------------------|------------|-------------|------------|-------------|-----------|------------|----------|-----------|
> | Sequential RL (GRPO on DAPO)         | 14.8 | 32.4 | 18.5 | 30.6 | 63.6 | 85.1 | 83.5 | 45.1 |
> | Sequential RL (GRPO on first GSM8K, then DAPO)         | 13.3 | 26.3 | 18.8 | 34.9 | 66.4 | 82.2 | 82.6 | 45.3 |
> | Improved Sequential RL (Parallel Thinking SFT + GRPO on DAPO) | 15.8 | 25.2 | 20.4 | 37.5 | 65.8 | 86.4 | 83.5 | 46.4 |
> | Parallel-R1-Seen       | 19.2 | 38.9 | 19.4 | 37.1 | 70.5 | 85.0 | 86.7 | 48.9 |
> | Parallel-R1-Unseen (S1) | 17.7 | 37.8 | 18.3 | 33.2 | 69.7 | 88.9 | 82.6 | 47.1 |
> | Parallel-R1-Unseen (S2) | 19.0 | 42.2 | 16.3 | 31.8 | 67.5 | 91.5 | 84.5 | 46.8 |

---

> > ### Comment · Reviewer_CufE · 2025-11-26
> >
> > > AIME’25: 1.3 → 5.5 (mean@16), 10.2 -> 24.6 (pass@16), ...
> >
> > The reviewer acknowledges the effort on fixing the mistakes in baseline evaluation.
> >
> > > For the scalability statement. Multiverse requires high-quality SFT data, fully solved trajectories for every problem. This leads to fundamentally limited scalability: each additional training example requires expensive teacher generation.
> >
> > As indicated in the response above, Parallel-R1 also requires an LLM in the data curation process (Parallel-R1 and Multiverse both use a Deepseek LLM), which poses similar constraints on scalability. In addition, Multiverse does not require fully correct reasoning and answers in the trajectories (indicated in the s1 paper, https://arxiv.org/abs/2501.19393, in Sec. 2), which further reduces the requirements on data curation.
> >
> > > We also respectfully clarify that our cold-start dataset is not designed to solve complex problems such as those in DAPO. Instead, its sole purpose is to teach the structural format of parallel thinking, so that RL can explore and learn the reasoning behaviors itself. Evaluating the cold-start data on DAPO is therefore misaligned with its intended function.
> >
> > The reviewer acknowledges the intention of the cold-start dataset for format learning. It would be valuable to demonstrate that the proposed format-learning cold-start does not lead to degraded reasoning ability on complex problems, as discussed above.
> >
> > > Thank you for your suggestions. To ensure a fully fair comparison, we trained the improved sequential reasoning baseline exactly following the reviewer’s protocol: Qwen3-4B-Base (without adding special tokens) → parallel-thinking SFT (treating special tokens as plain text) → GRPO on DAPO ( sequential inference engine).The results are presented below.
> >
> > Thanks for the additional experiment. This indeed resolves part of the reviewer's concern that the sequential reasoning baseline is not fair. The reviewer recommends putting the result in the main paper.

---

> > > ### Author Response · Authors · 2025-12-03
> > >
> > > >  As indicated in the response above, Parallel-R1 also requires an LLM in the data curation process (Parallel-R1 and Multiverse both use a Deepseek LLM), which poses similar constraints on scalability. In addition, Multiverse does not require fully correct reasoning and answers in the trajectories (indicated in the s1 paper, https://arxiv.org/abs/2501.19393, in Sec. 2), which further reduces the requirements on data curation.
> > >
> > > We have visualized the data pipeline of both Multiverse and Parallel-R1 below. We can see Parallel-R1 and Multiverse use LLMs with different purposes, e.g., transformation and generation, and require different capability-level LLMs, e.g., Gemini-2.5-Pro and Deepseek-Distiled-Qwen3-8B. Additionally, whether Multiverse requires or does not require fully correct reasoning and answers in the trajectories **does not affect the transformation stage**. This stage is costly and affects scalability.
> > >
> > >
> > >
> > >      +-------------------------------------------+
> > >      |      Generated Teacher Trajectory         |
> > >      |(raw reasoning chain from stronger models) |
> > >      |(**We agree this paratis easy to obtain**) |
> > >      +-------------------------------------------+
> > >                        |
> > >                        | +---------------------------------------------------------------+
> > >                        | |       Using Gemini-2.5-Pro (five step pipelines  |                        |
> > >                        | | Step 1: Parse the Chain into a Tree-structure Summary         |
> > >                        | | Step 2: Identify Parallel Nodes                               |
> > >                        | | Step 3: Reformat the Summary Tree into the MapReduce 	                                |           Structure via Control Tags                            |
> > >                        | | Step 4: Refill the Full, Detailed Reasoning Trajectories      |
> > >                        | | Step 5: Add Map and Reduce Stages, Rewrite All Paths          |
> > >                        V +---------------------------------------------------------------+
> > >      +-------------------------------------------+
> > >      |             Multiverse data               |
> > >      |   <Parallel>                              |
> > >      |      <Goal>                               |
> > >      |      <Outline> ... </Outline> ...         |
> > >      |      </Goal>                              |
> > >      |      <Path> ... </Path>                   |
> > >      |      <Path> ... </Path>                   |
> > >      |      <Conclusion> ... </Conclusion>       |
> > >      |   </Parallel>                             |
> > >      +-------------------------------------------+
> > >
> > >
> > >
> > >
> > >
> > >      +-------------------------------------------+
> > >      |                Question                   |
> > >      |                                           |
> > >      +-------------------------------------------+
> > >                        |
> > >                        | Deepseek-Qwen-3-8B
> > >      +-------------------------------------------+
> > >      |             Parallel-R1 data              |
> > >      |   <Parallel>                              |
> > >      |      <Path> ... </Path>                   |
> > >      |      <Path> ... </Path>                   |
> > >      |   </Parallel>                             |
> > >      |    <Summary>                              |
> > >      |     ...                                   |
> > >      |    </Summary>                             |
> > >      +-------------------------------------------+
> > >
> > >
> > >
> > >
> > >
> > >
> > > >  The reviewer acknowledges the intention of the cold-start dataset for format learning. It would be valuable to demonstrate that the proposed format-learning cold-start does not lead to degraded reasoning ability on complex problems, as discussed above.
> > >
> > > As discussed above, the question of whether the proposed format-learning cold-start preserves or degrades the reasoning ability of an already strong model does **not** align with the scientific goal of our work.
> > > We treat **parallel thinking as an independent reasoning paradigm**, rather than as a component to be inserted into an existing sequential reasoning model. Therefore, we want to learn parallel thinking behaviors from base model.
> > >
> > > Since our intention is **not** to build upon or modify strong sequential reasoning models, evaluating whether the cold-start stage affects the reasoning ability of such models falls outside the scope of our study.
> > >
> > >
> > >
> > > > Thanks for the additional experiment. This indeed resolves part of the reviewer's concern that the sequential reasoning baseline is not fair. The reviewer recommends putting the result in the main paper.
> > >
> > > We have put the results in the main paper.

---

> ### Author Response · Authors · 2025-11-26
> **Author Response (4/n)**
>
> > In Table 1, the "Parallel-R1-Seen", an autoregressive method (L342) that does not prevent paths from seeing each other in training, outperforms parallel methods (S1 and S2) by a large margin. Furthermore, there might be a discrepancy between training and inference, as the paths are generated in parallel at inference.
>
> Thank you for the thoughtful question. We clarify both points.
>
> **(1) Why Parallel-R1-Seen outperforms S1/S2 and strictly parallel-training variants.**
>
>  LLMs are pretrained entirely with the autoregressive next-token objective, which produces a strong inductive bias. When we enforce strict path isolation or non-AR parallel masks during post-training, the optimization objective becomes misaligned with this pretrained factorization. Given the small amount of post-training data, the model cannot reliably adapt to the new objective, which empirically leads to degraded performance.Parallel-R1-Seen preserves the autoregressive objective while using parallel inference during rollouts, achieving a better alignment with the pretrained model and therefore better performance.
>
> **(2) Whether there is a discrepancy between training and inference.**
>
> Although the optimization objective for training Parallel-R1-Seen is autoregressive, the rollout environment in RL uses the same parallel inference mechanism as test-time, meaning the model learns its parallel reasoning behavior under the same generation dynamics used at evaluation. The use of AR log-probabilities for policy gradient is the standard in GRPO and has minimal impact compared to the instability introduced by breaking the AR inductive bias from our empirical observation (as reflected by the performance gap between the Seen and Unseen variants).
>
> > The method is proposed to improve both performance and efficiency (L129, L300). However, no efficiency comparisons are provided with the baseline methods. The # Parallel is not an efficiency metric because the model can output much longer sequences in parallel to achieve the same performance as in sequential, causing higher latency than in sequential. A valid comparison metric is the latency in terms of the tokens, or wall-clock time if an efficient inference engine is used.
>
> **We did not try to use any metric to measure efficiency.** For example, in line 300,  we mentioned “We utilize Qwen-3-4B-Base (Yang et al., 2025a) as our backbone, which is the latest state-of-the-art open-source model at this scale, thus offering an ideal balance between performance and efficiency.” The sentence is to express why we choose the 4B model as baseline, but not to say training Parallel-R1 is more efficient than Multiverse or other SFT models.
>
> **We also did not state or imply lower latency, fewer generated tokens, or faster inference.** Mentions of parallelism in the introduction and related work refer only to the general potential of parallel thinking or to prior work, not to an efficiency advantage of our method. The “#Parallel” ratio in Table 2 is purely a behavioral diagnostic showing how often the learned policy activates parallel thinking, rather than an inference-efficiency metric. Our actual contribution is algorithmic: we present the first RL framework that systematically instills parallel reasoning for general reasoning tasks,, along with analyses of reward design, curriculum, and training dynamics.
>
> **We have revised the L129 and L300 now to avoid unintended interpretations.**
>
> > Writing is not clear: abstract and intro mentions a 42.9% improvement over sequential RL baselines, but this number is not reported in the experiments.
>
> Thank you for raising this concern. 42.9% improvement over sequential RL baselines is drawn from our mid-training experiments. The peak of our approach (red line) is 25.6% while the peak of baseline (gray line) is 17.917, so the relative improvement is 42.9% (25.6-17.917/17.917). We will add this improvement in our mid-training experiments.

---

> > ### Comment · Reviewer_CufE · 2025-11-26
> >
> > > Given the small amount of post-training data, the model cannot reliably adapt to the new objective, which empirically leads to degraded performance. Parallel-R1-Seen preserves the autoregressive objective while using parallel inference during rollouts, achieving a better alignment with the pretrained model and therefore better performance.
> >
> > Thanks for the clarification. The acknowledgement that the model cannot reliably adapt to parallel thinking is valuable and interesting. However, it sort of undermines the claim that Parallel-R1's SFT dataset is designed to learn the format of parallel thinking.
> >
> > Since the authors indicate that the post-training data is not enough, is it possible to demonstrate improved performance with more data, which could be obtained through the data pipeline? If the addition of data does not lead to improved performance, this would weaken the design choice of letting the model learn only the format in SFT.
> >
> > > Although the optimization objective for training Parallel-R1-Seen is autoregressive, the rollout environment in RL uses the same parallel inference mechanism as test-time, meaning the model learns its parallel reasoning behavior under the same generation dynamics used at evaluation.
> >
> > Thanks for acknowledging the discrepancies between the training and evaluation pipeline. The reviewer has a question: what is the difference between the ablation (Improved Sequential RL (Parallel Thinking SFT + GRPO on DAPO)) and Parallel-R1-Seen?
> >
> > A clarification would be valuable to help the reviewer understand the difference between the two.
> >
> > > We did not try to use any metric to measure efficiency.
> > > We also did not state or imply lower latency, fewer generated tokens, or faster inference.
> > > We have revised the L129 and L300 now to avoid unintended interpretations.
> >
> > The revision is appreciated. It is now clear that the authors did not claim efficiency as a key feature of Parallel-R1.
> >
> > Since the performance is the key metric, the reviewer believes that the authors should provide more evidence to support the claim that Parallel-R1 is more efficient than the sequential baseline on long CoT problems (e.g., Polaris-4B).

---

> > > ### Author Response · Authors · 2025-12-03
> > >
> > > > Thanks for the clarification. The acknowledgement that the model cannot reliably adapt to parallel thinking is valuable and interesting. However, it sort of undermines the claim that Parallel-R1's SFT dataset is designed to learn the format of parallel thinking. Since the authors indicate that the post-training data is not enough, is it possible to demonstrate improved performance with more data, which could be obtained through the data pipeline? If the addition of data does not lead to improved performance, this would weaken the design choice of letting the model learn only the format in SFT.
> > >
> > > Thank you for the comment. Our statement that the model “cannot reliably adapt to the new objective” refers **only** to the architectural change in *Parallel-R1-Unseen* (explicit path-isolation), which conflicts with the autoregressive pre-training objective. With limited post-training data (SFT data + RL data compared to trillions of tokens during pre-training stage), the model cannot fully adapt to this **architectural deviation**, leading to the degraded performance of Unseen. This issue is **unrelated to the design or effectiveness of our SFT**. **An empirical evidence as reflected in Table 2, is that Parallel-R1-Seen keeps the autoregressive architecture intact, and also uses the same SFT cold start data that can work well.**
> > >
> > > **Regarding scaling the SFT data:** this would test a **different research question** (scaling parallel-thinking CoT). Since the goal of SFT is to teach model the parallel-thinking format and the model after SFT stage already can produce parallel thinking formats (Table 2, # Parallel metric), this means our current SFT stage works well. Once this objective is achieved, adding more SFT-format data does not relate to our design choice and is outside the scope of this work.
> > >
> > > > Thanks for acknowledging the discrepancies between the training and evaluation pipeline. The reviewer has a question: what is the difference between the ablation (Improved Sequential RL (Parallel Thinking SFT + GRPO on DAPO)) and Parallel-R1-Seen?
> > >
> > > Thank you for the question. The “Improved Sequential RL” baseline and “Parallel-R1-Seen” differ in **two essential aspects**,
> > >
> > > - 1) **Rollout Strategy is different:**
> > >      - The improved sequential baseline uses standard autoregressive rollouts, whereas Parallel-R1-Seen uses parallel-inference rollouts required for learning parallel-thinking behaviors.
> > >   2) **RL curriculum**:
> > >      - The improved sequential baseline follows the reviewer-suggested “GRPO on DAPO” setup.
> > >        Parallel-R1-Seen follows our parallel-thinking curriculum (GSM8K RL + DAPO RL) that matches its rollout dynamics. As shown in Table 2, GSM8K RL does not lead to measurable improvement on sequential baseline, so adding it to the improved sequential baseline would not change its outcome.
> > >
> > > These differences arise from the **policy being learned**, not differences in supervision strength.

---

> ### Author Response · Authors · 2025-11-26
> **Author Response (5/n)**
>
> > Writing is not clear: abstract and intro mentions a 42.9% improvement over sequential RL baselines, but this number is not reported in the experiments.
>
> Thank you for raising this concern. 42.9% improvement over sequential RL baselines is drawn from our mid-training experiments. The peak of our approach (red line) is 25.6% while the peak of baseline (gray line) is 17.917, so the relative improvement is 42.9% (25.6-17.917/17.917). We added the number in our mid-training section.
>
> > In Fig. 4, the parallel ratio goes to 0 starting at Step 300, but the parallel ratio is non-zero in the Parallel-R1 results in Table 2. In L308, the authors indicated that the model is trained with 300 gradient update steps in Stage 2 (which is around Step 500 in total in Fig. 4, as there is a 200-step RL training in Stage 1). These two results are inconsistent. Furthermore, if training for 300 steps leads to a parallel ratio of 0, it indicates that the model will not be able to perform parallel reasoning at inference and will not have improved efficiency compared to the sequential baseline.
>
> Thank you for the question. There is no inconsistency between Figure 4 and Table 2. These two results come from different training pipelines.
>
> Table 2 (Parallel-R1-Unseen S2) uses the main training recipe described in Section 3.4, where the model is trained only with the alternating reward scheme. Since the model continues to receive periodic incentives for parallel reasoning, it retains a non-zero parallel ratio at the checkpoints used for evaluation.
>
> Figure 4, by contrast, is an independent diagnostic experiment designed solely to analyze the role of parallel thinking as a mid-training exploration scaffold. It explicitly adds a separate accuracy-only phase, during which the parallel ratio naturally decays toward zero. This experiment is not used to obtain any results in Table 2.
>
>
> > What are the training time and hardware requirements for Parallel-R1-Unseen (4B)? How does it compare to Multiverse-32B?
>
> Parallel-R1-Unseen (4B) requires 8 x 40g gpu and will need to train around 5-6 days. According to Multiverse paper, Multiverse-32B requires 3 hours on 8 NVIDIA B200 GPUs.
>
> We would like to note that the overall cost profile of Multiverse is influenced not only by the model-training phase but also by the data-construction pipeline. Multiverse relies on high-quality parallel-reasoning demonstrations generated by frontier LLMs (e.g., Gemini-2.5-Pro), and in our attempts to reproduce this pipeline, we found that the cost per generated sample can be quite high—often several dollars per example.
>
> > What is the exact inference setting for "Parallel-R1-Seen"? Is it with a parallel inference engine?
>
> Yes, it is with a parallel inference engine.
> > Are there any interpretations for going fully sequential after 300 steps in Fig. 4?
>
> The transition to fully sequential behavior in Fig. 4 is expected, because this figure corresponds to the experiment in Section 4.5, where the training switches to an accuracy-only reward phase. At this phase, there is no distinction between using parallel and not using parallel. Once the reward no longer provides any benefit for using parallel reasoning, the model naturally converges to the safer and more stable sequential strategy under this objective, causing the parallel ratio to decay to zero. This phenomenon will not happen in our Parallel-r1-Unseen (S2).

---

> > ### Comment · Reviewer_CufE · 2025-11-26
> >
> > Thanks for the clarification and revision of the paper to address the comments! This really makes the paper more rigorous and clearer.
> >
> > The reviewer has one more question and one more concern after going through the rebuttal:
> > 1. How does the average sequence length on AIME 24/25 compared to Qwen3-4B? This would provide more insights into the test-time scaling beahvior from the proposed training pipeline.
> > 2. The reviewer is still concerned about the setting. As mentioned in DeepSeek-R1 paper (Table 6), running RL without SFT is much worse than running SFT only (which is worse than running RL after SFT, as justified in DeepScaleR/RLLM and Polaris) on small models. This is why DeepSeek, Qwen3, and DAPO only run RL on their largest models, and not much effort is put into training smaller models with DAPO without cold-start. Since training pipelines such DAPO are not designed for small models, it could lead to insufficient test-time scaling behavior, potentially leading to much shorter CoTs. Could the authors provide more evidence to support rationale behind the choice of DAPO on a small model on parallel thinking, since it's demonstrated to be relatively worse for training sequential reasoning?

---

> ### Author Response · Authors · 2025-11-26
> **Thank you for your review.**
>
> Thanks for your detailed reviews. We really appreciate them! We've incorporated your advice to revise our PDFs, removing statements that might cause confusion.
>
> We also want to emphasize that Parallel-R1 is novel and clearly distinct from Multiverse. Parallel-R1 is an RL pipeline, whereas Multiverse is an SFT pipeline. In fact, RL training can be applied on top of Multiverse’s SFT pipeline, forming a natural progression: Base model → SFT model (Multiverse) → RL-trained model (Parallel-R1).
>
> This approach aligns with many recent LLM post-training practices, such as those used in Qwen and DeepSeek, where applying RL after SFT typically yields further gains. RL enables greater exploration and on-policy optimization, which complements the SFT stage.

---

> > ### Comment · Reviewer_CufE · 2025-11-26
> >
> > Thanks very much as well for the detailed responses and for updating the paper to address the comments! While some concerns remain, the reviewer sincerely appreciates the time and effort the authors have put into addressing the comments.
> >
> > Since the rebuttal mentions that "applying RL after SFT typically yields further gains" several times, the reviewer would like to ask the authors to provide more evidence to support the claim by comparing their data pipeline with Polaris-4B to see if applying the format learning SFT + parallel RL can indeed compete with open RL recipies based on SFT.
> >
> > Although the reviewer has raised many concerns, the intention is to foster a healthy discussion and improve the paper. The reviewer would be happy to recommend the paper if these issues can be convincingly addressed in the final version.

---

> ### Author Response · Authors · 2025-12-02
> **Further Clarification on Difference between Parallel-R1 and Multiverse (2/n)**
>
> ## **5. On the point using trajectories from Deepseek or other open reasoning models is not a technical difficulty.**
>
>
>
>      +-------------------------------------------+
>      |      Generated Teacher Trajectory         |
>      |(raw reasoning chain from stronger models) |
>      |(**We agree this paratis easy to obtain**) |
>      +-------------------------------------------+
>                        |
>                        | +---------------------------------------------------------------+
>                        | |       Using Gemini-2.5-Pro (five step pipelines  |                        |
>                        | | Step 1: Parse the Chain into a Tree-structure Summary         |
>                        | | Step 2: Identify Parallel Nodes                               |
>                        | | Step 3: Reformat the Summary Tree into the MapReduce 	                                |                                     Structure via Control Tags                            |
>                        | | Step 4: Refill the Full, Detailed Reasoning Trajectories      |
>                        | | Step 5: Add Map and Reduce Stages, Rewrite All Paths          |
>                        V +---------------------------------------------------------------+
>      +-------------------------------------------+
>      |             Multiverse data               |
>      |   <Parallel>                              |
>      |      <Goal>                               |
>      |      <Outline> ... </Outline> ...         |
>      |      </Goal>                              |
>      |      <Path> ... </Path>                   |
>      |      <Path> ... </Path>                   |
>      |      <Conclusion> ... </Conclusion>       |
>      |   </Parallel>                             |
>      +-------------------------------------------+
>
> We acknowledge that abundant raw reasoning trajectories from Deepseek and other open models are readily available. However, **transforming these raw trajectories into high-quality parallel-thinking data (as required by Multiverse) requires a multi-stage semantic transformation pipeline (e.g., parsing, restructuring, node identification, MapReduce-style rewriting, and trajectory refilling).**
> This step, rather than obtaining the raw trajectories themselves, is the costly and technically involved component.
>
> ## **6. On the orthogonal contribution between Multiverse and Parallel-R1**
>
> **The reviewer’s claim is based on a misunderstanding of what we mean by “orthogonal.” We use “orthogonal” to refer to the learning paradigms (imitation vs. RL exploration), not to the ability to combine two data pipelines.**
>
> Even if future work studies a *new* setting that combines Multiverse-style SFT with the RL components (reward modeling) of Parallel-R1, that corresponds to a **different research problem**, and does not affect the methodological distinction we draw in this paper.
>
> ## **Summary**
>
> Parallel-R1 and Multiverse represent **two complementary paradigms** for studying parallel thinking:
>
> - **Multiverse:** imitation learning → transferring teacher-provided parallel reasoning
> - **Parallel-R1:** RL exploration → examining whether parallel reasoning can be learned through exploration
>
> Our cold-start pipeline is deliberately aligned with the latter objective and ensures that RL can effectively explore parallel thinking behaviors. Using Multiverse-style SFT would correspond to a *different research setting*, not the one investigated in this work.

---

> ### Author Response · Authors · 2025-12-03
>
> > How does the average sequence length on AIME 24/25 compared to Qwen3-4B? This would provide more insights into the test-time scaling beahvior from the proposed training pipeline.
>
> As requested, we report the average output lengths on AIME24/25.
>
> First, we directly address the comparison with the official Qwen3-4B model:
>  **Qwen3-4B typically produces much longer chains**, because it is trained with large-scale long-trajectory distillation. In contrast, **Parallel-R1 is trained under a standard research setting with a maximum training length of 3k tokens[1]**, so its inference length naturally remains much shorter. This primarily reflects different training budgets rather than an inherent limitation of our method.
>
> Because these settings are fundamentally different, the meaningful apples-to-apples comparison is **Qwen3-4B-Base + GRPO**, which uses the *same* length constraints as Parallel-R1.
>
> **Average output length (AIME24/25):**
>
> - **Qwen3-4B:** 16213
> - **Qwen3-4B-Base + GRPO:** 1437
> - **Parallel-R1-Unseen (S2):** 1836
>
> Parallel-R1 produces longer outputs than the GRPO baseline.
>
> [1] TTRL: Test-Time Reinforcement Learning.
>
> > The reviewer is still concerned about the setting. As mentioned in DeepSeek-R1 paper (Table 6), running RL without SFT is much worse than running SFT only (which is worse than running RL after SFT, as justified in DeepScaleR/RLLM and Polaris) on small models. This is why DeepSeek, Qwen3, and DAPO only run RL on their largest models, and not much effort is put into training smaller models with DAPO without cold-start. Since training pipelines such DAPO are not designed for small models, it could lead to insufficient test-time scaling behavior, potentially leading to much shorter CoTs. Could the authors provide more evidence to support rationale behind the choice of DAPO on a small model on parallel thinking, since it's demonstrated to be relatively worse for training sequential reasoning?
>
>
> DAPO-based RL pipelines on small models are the standard experimental setting in recent RL-for-reasoning studies [1–6]. We follow this widely adopted benchmark setup to ensure consistency and comparability with existing work.
>
> [1] Beyond the 80/20 rule: High-entropy minority tokens drive effective reinforcement learning for llm reasoning, Neurips 2025.
>
> [2] Explore Data Left Behind in Reinforcement Learning for Reasoning Language Models
>
> [3] FlowRL: Matching Reward Distributions for LLM Reasoning
>
> [4] Reinforcement learning with verifiable rewards implicitly incentivizes correct reasoning in base llms
>
> [5] Knapsack rl: Unlocking exploration of llms via optimizing budget allocation
>
> [6] Understanding Tool-Integrated Reasoning

---

### Official Review · Reviewer_qZmU · 2025-11-04

**Soundness:** 3
**Presentation:** 3
**Contribution:** 3
**Rating:** 6
**Confidence:** 3

**Summary:**

This paper introduces Parallel-R1, which uses RL to teach LLMs to perform "parallel thinking" (exploring multiple reasoning paths concurrently) for complex mathematical problems.

To overcome the difficulty of training this behavior from scratch (the "cold-start" problem), the framework uses a progressive curriculum: the model first learns the basic format of parallel thinking on easier math problems. Then, the model then uses RL to generalize this skill to more difficult problems, guided by a reward system that balances task accuracy with the use of parallel structures.

The authors show that Parallel-R1 improves accuracy on challenging math benchmarks (MATH, AIME) compared to standard sequential RL baselines. The model's strategy evolves during training, shifting from using parallel thinking for broad exploration in early stages to using it for multi-perspective verification in later stages.

In addition, this paper claims that Parallel thinking can serve as a "mid-training exploration scaffold," where an initial phase of forced parallel exploration helps the model discover better strategies, leading to a higher final performance ceiling even after the model reduces its use of parallel structures.

**Strengths:**

- As far as I know, this is the first RL framework to teach parallel thinking for complex, real-world mathematical reasoning tasks, moving beyond prior work that focused on synthetic domains or supervised fine-tuning.
- The paper presents a simple and scalable data pipeline for generating the initial training data, which avoids the computationally intensive and complex pipelines required by previous methods.
- The proposed framework demonstrates significant and consistent performance improvements across multiple challenging mathematical reasoning benchmarks, including MATH, AIME, and AMC.
- The research proposes a novel concept of using parallel thinking as a "mid-training exploration scaffold," showing that this intermediate phase helps the model discover better policies and achieve a higher final performance.

**Weaknesses:**

- The entire methodology is validated exclusively on mathematical reasoning tasks. The paper makes no attempt to demonstrate or even discuss how this "parallel thinking" capability would transfer to more ambiguous, open-ended, or creative domains like writing, where the notion of distinct, verifiable "paths" is far less clear.

- The paper claims that parallel thinking can be achieved at "negligible cost" by exploiting GPU parallelism. This is an oversimplification. Generating multiple reasoning trajectories simultaneously consumes substantially more memory and computational resources (total FLOPs) than a single path. While wall-clock time might be similar, it ignores the increased hardware requirements and energy consumption.

- The intriguing idea of parallel thinking as a "mid-training scaffold" is supported by a single experiment on one benchmark (AIME25) with an arbitrarily chosen cutoff point (200 steps). This is preliminary evidence at best.

- The most effective reward scheme for the structured model involves a finely-tuned "alternating" schedule (80% accuracy, 20% parallel reward, window of 10 steps). This feels more like ad-hoc hyperparameter tuning than a principled approach, adding significant complexity to the training process.

**Questions:**

See weaknesses.

---

> ### Author Response · Authors · 2025-12-02
> **Author response (1/n)**
>
> We thank the reviewer for the constructive and thoughtful comments. The feedback has helped us refine our presentation and better clarify the contributions and limitations of the work. We address each point in detail below.
>
> > The entire methodology is validated exclusively on mathematical reasoning tasks. The paper makes no attempt to demonstrate or even discuss how this "parallel thinking" capability would transfer to more ambiguous, open-ended, or creative domains like writing, where the notion of distinct, verifiable "paths" is far less clear.
>
> We thank the reviewer for this constructive feedback. We fully acknowledge that the exclusive focus on mathematical reasoning tasks is a key limitation of the current validation. We have added it in the Limitation section.
>
> We chose mathematical reasoning as our primary validation domain because it provides discrete and certain steps and objective standards for correctness. This highly structured environment is beneficial for first proving the concept whether RL can learn parallel thinking beyond synthetic tasks, e.g., general mathematical reasoning tasks.
>
> While our experiments focus on math, we view parallel thinking as a potentially general mechanism. For instance, in open-ended writing, one could imagine LLMs exploring multiple drafts or perspectives before producing a final summary. We emphasize that this remains a hypothesis rather than a claim of demonstrated capability in the present work.
>
> As a preliminary probe of generality beyond math, we evaluated a Parallel-R1 variant (trained exclusively on math) on a non-math reasoning dataset, StrategyQA. Even without fine-tuning on the target domain, Parallel-R1 outperforms the GRPO baseline, suggesting that certain aspects of parallel-thinking behavior may transfer across tasks. However, evaluating open-ended creative domains remains important future work.
>
>
> | Method             | StrategyQA Accuracy |
> |--------------------|---------------------------|
> | GPRO (Baseline)    |        85.7          |
> | Parallel-R1-Seen   |         89.7 (+4.0)  |
> | Parallel-R1-Unseen (S1) |       86.8 (+1.1)            |
> | Parallel-R1-Unseen (S2) |       86.6 (+0.9)            |
>
>
>
>
> > The paper claims that parallel thinking can be achieved at "negligible cost" by exploiting GPU parallelism. This is an oversimplification. Generating multiple reasoning trajectories simultaneously consumes substantially more memory and computational resources (total FLOPs) than a single path. While wall-clock time might be similar, it ignores the increased hardware requirements and energy consumption.
>
> Thank you for this insightful comment. We agree that the term "negligible cost" in the introduction was an oversimplification and led to ambiguity regarding resource consumption. The reviewer is correct that generating multiple reasoning trajectories simultaneously consumes substantially more memory and computational resources (total FLOPs) than a single path.
>
> To avoid misunderstanding, we will revise the sentence as follows:
> Moreover, parallel thinking provides such benefits by leveraging the concurrent execution capability of modern GPUs to explore multiple reasoning paths in parallel [1].
>
> This clarification concerns only introductory wording and does not affect the method or claims of the paper.
>
> [1] Multiverse: Your Language Models Secretly Decide How to Parallelize and Merge Generation.

---

> > ### Author Response · Authors · 2025-12-02
> > **Author response (2/n)**
> >
> > > The intriguing idea of parallel thinking as a "mid-training scaffold" is supported by a single experiment on one benchmark (AIME25) with an arbitrarily chosen cutoff point (200 steps). This is preliminary evidence at best.
> >
> > Thank you for pointing this out. We agree this analysis is exploratory and is not positioned as a core contribution (as reflected in the “extra bonus” section title). We have revised Sec.4.6 to make this framing even clearer.
> >
> > **Regarding the choice of the 200-step cutoff:** The cutoff was not selected arbitrarily. As shown in Figure 3, the model’s parallel-structure behavior exhibits a clear transition in the near 200 steps, where it enters its first stable plateau. We therefore used a 200-step cutoff as the natural switching point for the scaffold experiment.
> >
> > To verify this is not sensitive to the exact checkpoint, we select 210 steps checkpoint and train additional 100 steps via accuracy-only reward. We can find similar observations as that shown in Figure 4: 1) parallel ratio decreases, and 2) then performance begins to increase. Although we could not afford to continue training for the full 300 steps due to rebuttal-time/GPU constraints, the early-stage replication of the key trend provides supportive evidence that the mid-training scaffold is not tied to a single cutoff point.
> >
> > **On using only AIME25:** We choose AIME25 because this is the most challenging benchmark among the AIME24/AIME25/AMC/MATH, where mid-training scaffolding is most visible. However, to directly address the reviewer’s concern, we additionally report benchmark results on AMC, and we observe the same transition pattern (parallel ratio decreases → accuracy rises). We include the results below.
> >
> > | Metric                  | Value Range (200–600 steps) | Trend                        |
> > |-------------------------|------------------------------|------------------------------|
> > | Parallel Ratio (%)      | 62.3 → ~0                      | **Parallel Ratio decreasing to 0**  |
> > | AMC23 Pass@1 Accuracy (%)     | 63.1 → 71.3  >  67.8 (Baseline peak within same steps)               | **AMC performance increases**|
> >
> > > The most effective reward scheme for the structured model involves a finely-tuned "alternating" schedule (80% accuracy, 20% parallel reward, window of 10 steps). This feels more like ad-hoc hyperparameter tuning than a principled approach, adding significant complexity to the training process.
> >
> > Thanks for raising this concern. First, we would like to clarify that the alternating reward scheme is not ad-hoc. It directly addresses a practical conflict between two objectives: (1) solving the problem correctly and (2) producing meaningful parallel structures. When optimized simultaneously, the model often collapses to correct-but-non-parallel or syntactic-but-meaningless parallel outputs. The alternating schedule is a minimal way to decouple these signals so that accuracy can stabilize before structure is reinforced. This requires only a few lines of code.
> >
> > We also tested multiple weights, window sizes, and frequencies. All variants outperform the GRPO baseline on average, indicating that the scheme is robust rather than hyperparameter-fragile. We have added the results in Appendix F.
> >
> >
> > | Setting                                | AIME24 | AIME25 | AMC23 | MATH |  Avg. |
> > |----------------------------------------|--------|--------|--------|-------|-------|
> > | GRPO (DAPO)             |      18.5    |      14.8    |     63.6    |    83.5   |  45.1  |
> > | weight = 1.2 (Parallel-R1-Unseen (S2))              |      16.3    |     19.0    |    67.5    |    84.5   |  46.8  |
> > | weight = 2.0 (Stronger struct. bias) |   17.7     |    17.9    |    64.5    |  84.2     | 46.1  |
> > | Window-size = 10 (Parallel-R1-Unseen (S2))        |    16.3    |     19.0    |    67.5    |    84.5   |  46.8 |
> > | Window-size = 20              |   17.5     |   19.4    |    65.3    |   84.2    |  46.6 |
> > | Frequency = 8/2 (Parallel-R1-Unseen (S2)) |    16.3    |     19.0    |    67.5    |    84.5   |  46.8 |
> > | Frequency = 6/4        |   18.2     |    19.6    |   65.6     |   83.9    | 46.8  |
> >
> > **We hope the above clarifications and additional analyses fully address the reviewer’s concerns.**

---

### Official Review · Reviewer_HXbg · 2025-11-05

**Soundness:** 4
**Presentation:** 4
**Contribution:** 4
**Rating:** 8
**Confidence:** 2

**Summary:**

This paper proposes Parallel-R1, the first Reinforcement Learning framework designed to instill parallel thinking in large language models for complex reasoning tasks. Parallel thinking allows the model to explore multiple reasoning paths concurrently, which is hypothesized to improve reasoning generalization. The key idea is a progressive curriculum: first, the model is trained on simpler tasks via supervised fine-tuning (SFT) to bootstrap parallel thinking behavior, and then RL is applied on more difficult tasks to generalize this skill.

The authors explore multiple reward schemes, including an alternating reward strategy that balances outcome-based accuracy with parallel thinking behaviors. They also introduce two structural variants—Parallel-Seen (no architectural changes) and Parallel-Unseen (incorporates inductive biases in the attention mechanism to enforce path isolation). Experiments on a suite of math benchmarks (MATH, AMC23, AIME24/25) demonstrate substantial gains: Parallel-R1 improves accuracy by 8.4% over sequential RL models and achieves a 42.9% improvement when parallel thinking is treated as a mid-training exploration scaffold. The paper also provides in-depth analyses of learning dynamics and ablation studies, showing that parallel thinking evolves from exploration to verification during training.

**Strengths:**

The paper is highly novel, introducing the first RL framework explicitly designed to train parallel thinking in LLMs. The progressive curriculum that bootstraps learning from simple tasks to more complex problems is well-motivated and effectively addresses the cold-start problem in RL. The experimental results are strong and demonstrate substantial improvements on multiple established math benchmarks, including MATH, AMC23, and AIME24/25. Beyond empirical performance, the paper provides insightful analyses of learning dynamics, revealing that parallel thinking initially functions as an exploration mechanism and later shifts toward verification. The methodological rigor is evident in the extensive ablation studies, careful examination of reward schemes, and exploration of architectural variants. Furthermore, the framework achieves these gains while requiring minimal modifications to the model architecture, which enhances its practical applicability.

**Weaknesses:**

Despite the strong contributions, there are several areas where clarification or further investigation would strengthen the work. First, the generality of Parallel-R1 beyond mathematical reasoning is unclear, and it would be valuable to know whether similar gains could be achieved in other complex reasoning domains, such as commonsense or scientific reasoning. Second, the computational cost of RL on LLMs is potentially significant, yet the paper provides limited discussion of efficiency or resource requirements compared to sequential RL or SFT-only approaches.

**Questions:**

The paper demonstrates strong results on mathematical reasoning benchmarks, but it is unclear how well the Parallel-R1 framework would generalize to other domains requiring complex reasoning, such as commonsense reasoning or scientific problem-solving. Can the authors comment on the expected transferability of parallel thinking beyond math tasks?

---

> ### Author Response · Authors · 2025-11-27
> **Author response**
>
> Thank you for taking time to review our paper and for the valuable feedback. We believe these feedback indeed help improve our paper a lot. We are glad to address your questions point by point below.
>
> > First, the generality of Parallel-R1 beyond mathematical reasoning is unclear, and it would be valuable to know whether similar gains could be achieved in other complex reasoning domains, such as commonsense or scientific reasoning. & Can the authors comment on the expected transferability of parallel thinking beyond math tasks?
>
> Thanks for your suggestion. To further examine the generality of our method, we additionally evaluated Parallel-R1 on the StrategyQA. The results are shown below. These findings provide preliminary evidence that the Parallel-R1 learned from mathematical reasoning can also transfer to non-mathematical domains, indicating that Parallel-R1 is not limited to math reasoning tasks. We have added these in revised version of our paper.
>
>
> | Method             | StrategyQA Accuracy |
> |--------------------|---------------------------|
> | GPRO (Baseline)    |        85.7          |
> | Parallel-R1-Seen   |         89.7 (+4.0)  |
> | Parallel-R1-Unseen (S1) |       86.8 (+1.1)            |
> | Parallel-R1-Unseen (S2) |       86.6 (+0.9)            |
>
> > Second, the computational cost of RL on LLMs is potentially significant, yet the paper provides limited discussion of efficiency or resource requirements compared to sequential RL or SFT-only approaches
>
> Thanks for your suggestion. Below we provide details of resource requirements and training time of our parallel-R1.
>
> **Training Efficiency.**
> We clarify the training-time comparison. The Parallel-R1-Seen has the comparable training cost as the sequential baseline: \~3.5 days on 8×40GB GPUs.  The Parallel-R1-Unseen variant requires a longer runtime (~6 days on the same hardware) because it must dynamically construct multiverse-style 4D attention masks at every RL rollout step. Unlike Multiverse, which preprocesses all 4D masks offline during supervised data construction, our RL setting must build these masks online for every trajectory, resulting in additional overhead. According to Multiverse paper, Multiverse-32B requires 3 hours on 8 NVIDIA B200 GPUs. We have added these in revised version of our paper.
>
> **We hope the above clarifications and additional analyses fully address the reviewer’s concerns.**

---

### Author Response · Authors · 2025-11-26
**Overall AC Summary Letter**

Dear reviewers and ACs,

Thank you for your constructive feedback. We appreciate that reviewers consistently highlighted several core strengths of the paper:

- **Novel and significant research question** of whether RL can induce parallel thinking (RoLa, 6K5f)
- **Clear motivation and principled method design** (HXbg, EJPr, RoLa)
- **Introduction of a novel RL framework and parallel-thinking mechanism** (qZmU, HXbg, EJPr, 6K5f, RoLa, CufE)
- **Strong empirical results and insightful analyses** across mathematical reasoning tasks (HXbg, qZmU, EJPr, RoLa)
- **Clarity/completeness of the overall pipeline** (EJPr, RoLa,CufE)

During the rebuttal period, we provided detailed clarifications and conducted substantial additional analyses to address raised concerns. Our updates fall into two major categories:

## 1. Added new experiments directly addressing reviewer concerns

- Path diversity analysis to measure quality of learned parallel thinking
- Reward sensitivity ablations: weight, window size, alternating ratio
- Analysis of transition phenomenon (“exploration → verification”) across larger models and non-math dataset
- High-variance analysis across dataset
- Robustness analysis of checkpoint selection in mid-training experiments
- Generalization beyond math: StrategyQA
- Skip-SFT ablation
- Larger model experiment: **Qwen-2.5-7B-Math**
- Multiverse-4B baseline
- Improved sequential baseline (Qwen3-4B-Base → parallel-format SFT → GRPO)
- Comparison with test-time parallel methods, including Self-Consistency
- Training cost comparison
- Added AMC mid-training results

## 2. Manuscript improvements

- Revised Sections **3.2**, **4.5**, and **4.6** for clarity and completeness
- Rephrased “negligible cost” in the Introduction to avoid misunderstanding
- Added a **Limitations and Future Direction** section

Because we received seven reviews and RL experiments take time to run, our rebuttal began to post on Nov 26. While most reviewers did not provide follow-up comments—likely due to time constraints—we ensured that **every concern was addressed** with new experiments, analyses, and manuscript revisions.

Thank you again to all reviewers, ACs, and PCs for your time, feedback, and engagement throughout the process.

---

### Meta-Review · Area_Chair_Zey4 · 2025-12-13

**Summary:**

The idea of applying RL to induce parallel thinking for LLMs is novel. In general, the experiments are comprehensive to demonstrate the effectiveness of the proposed method. One major concern about the lack of comparison between the proposed method and other parallel thinking approaches, such as Multiverse, is more or less addressed in the rebuttal. Another major concern about the lack of experiments on larger LLMs is also addressed in the rebuttal.

**Reviewer Concerns:**

Based on the in-depth discussions in the rebuttal, the majority of the reviewers' concerns have been adequately addressed. Reviewer CufE has offered some minor suggestions for further testing the performance of the proposed idea, but no major unresolved issues remain.

**Reviewer Scores:**

Initially, only reviewers iyKL (4) and CufE (2) recommended rejection. Reviewer iyKL’s concerns were non-critical and have been addressed in the rebuttal, though he/she did not participate in the discussion during the rebuttal. Reviewer CufE engaged in a thorough discussion with the authors. Most of his/her concerns have been resolved, and reviewer CufE would likely raise his score to 6 (estimate).

---

### Decision · Program_Chairs · 2026-01-26

Accept (Poster)